# CHARACTERISTIC NEURAL ORDINARY DIFFERENTIAL EQUATIONS

**Xingzi Xu** [1, *†], **Ali Hasan** [1,*], **Khalil Elkhalil**[1], **Jie Ding**[2], **Vahid Tarokh**[1]

[1] Department of Electrical and Computer Engineering, Duke University
[2] School of Statistics, University of Minnesota
`{xingzi.xu, ali.hasan, khalil.elkhalil, vahid.tarokh}@duke.edu`
`dingj@umn.edu`

## ABSTRACT

We propose Characteristic-Neural Ordinary Differential Equations (C-NODEs), a framework for extending Neural Ordinary Differential Equations (NODEs) beyond ODEs. While NODE models the evolution of latent variables as the solution to an ODE, C-NODE models the evolution of the latent variables as the solution of a family of first-order partial differential equations (PDEs) along curves on which the PDEs reduce to ODEs, referred to as characteristic curves. This reduction along characteristic curves allows for analyzing PDEs through standard techniques used for ODEs, in particular the adjoint sensitivity method. We also derive C-NODE-based continuous normalizing flows, which describe the density evolution of latent variables along multiple dimensions. Empirical results demonstrate the improvements provided by the proposed method for irregularly sampled time series prediction on MuJoCo, PhysioNet, and Human Activity datasets and classification and density estimation on CIFAR-10, SVHN, and MNIST datasets given a similar computational budget as the existing NODE methods. The results also provide empirical evidence that the learned curves improve the system efficiency using a lower number of parameters and function evaluations compared with those of the baselines.

## 1 INTRODUCTION

Deep learning and differential equations share many connections, and techniques in the intersection have led to insights in both fields. One predominant connection is based on certain neural network architectures resembling numerical integration schemes, leading to the development of Neural Ordinary Differential Equations (NODEs) (Chen et al., 2019b). NODEs use a neural network parameterization of an ODE to learn a mapping from observed variables to a latent variable that is the solution to the learned ODE. A central benefit of NODEs is the constant memory cost, when backward passes are computed using the adjoint sensitivity method rather than backpropagating through individual forward solver steps. Backpropagating through adaptive differential equation solvers to train NODEs will often result in extensive memory use, as mentioned in Chen et al. (2019b). Moreover, NODEs provide a flexible probability density representation often referred to as *continuous normalizing flows* (CNFs). However, since NODEs can only represent solutions to ODEs, the class of functions is somewhat limited and may not apply to more general problems that do not have smooth and one-to-one mappings. To address this limitation, a series of analyses based on methods from differential equations have been employed to enhance the representation capabilities of NODEs, such as the theory of controlled differential equations (Kidger et al., 2020), learning higher-order ODEs (Massaroli et al., 2021), augmenting dynamics (Dupont et al., 2019), and considering dynamics with delay terms (Zhu et al., 2021). Additionally, certain works consider generalizing the ODE case to partial differential equations (PDEs), such as in Ruthotto & Haber (2020) and Sun et al. (2019). These PDE-based methods do not use the adjoint method, removing the primary advantage of constant memory cost. This leads us to the central question motivating the work: can we combine the benefits

---

*Equal contribution.
†Corresponding Author.

of the rich function class of PDEs with the efficiency of the adjoint method? To do so, we propose a method of continuous-depth neural networks that solves a PDE over parametric curves that reduce the PDE to an ODE. Such curves are known as *characteristics*, and they define the solution of the PDE in terms of an ODE (Griffiths et al., 2015). The proposed Characteristic Neural Ordinary Differential Equations (C-NODE) learn both the characteristics and the ODE along the characteristics to solve the PDE over the data space. This allows for a richer class of models while still incorporating the same memory efficiency of the adjoint method. The proposed C-NODE is also an extension of existing methods, as it improves the empirical accuracy of these methods in classification tasks, time series prediction tasks, and image quality in generation tasks.

## 2 RELATED WORK

NODE is often motivated as a continuous form of a Residual Network (ResNet) (He et al., 2015), since the ResNet can be interpreted as a forward Euler integration scheme on the latent state (Weinan, 2017). Specifically, a ResNet is composed of multiple blocks with each block can be represented as: $u_{t+1} = u_t + f(u_t, \theta)$, where $u_t$ is the evolving hidden state at time $t$ and $f(u_t, \theta)$ is interpreted as the gradient at time $t$, namely $du/dt(u_t)$. Generalizing the model to a step size given by $\Delta t$ results in $u_{t+\Delta t} = u_t + f(u_t, \theta)\Delta t$. To adapt this model to a continuous setting, we let $\Delta t \to 0$ and obtain: $\lim_{\Delta t \to 0} (u_{t+\Delta t} - u_t)/\Delta t = du(t)/dt$. The model can then be evaluated through existing numerical integration techniques, as proposed by Chen et al. (2019b):

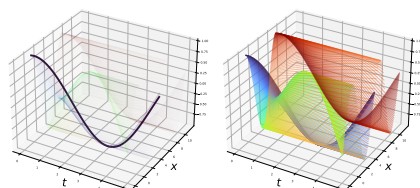

Single Characteristic (ODE)  Multiple Characteristics (PDE)

$$u(t_1) = u(t_0) + \int_{t_0}^{t_1} \frac{du(t)}{dt} dt = u(t_0) + \int_{t_0}^{t_1} f(u(t), t, \theta) dt.$$

Numerical integration can then be treated as a black box, using numerical schemes beyond the forward Euler to achieve higher numerical precision. However, since black box integrators can take an arbitrary number of intermediate steps, backpropagating through individual steps would require too much memory since the individual steps must be saved. Chen et al. (2019b) addressed this problem by using adjoint backpropagation, which has a constant memory usage. For a given loss function on the terminal state $t = 1$ of the hidden state $\mathcal{L}(u(t_1))$, the adjoint $a(t)$ is governed by another ODE:

Figure 1: Comparison of traditional NODE (left) and proposed C-NODE (right). The solution to NODE is the solution to a single ODE, whereas C-NODE represents a series of ODEs that form the solution to a PDE. Each color in C-NODE represents the solution to an ODE with a different initial condition. NODE represents a single ODE, and can only represent $u(x, t)$ along one dimension, for example, $u(x = 0, t)$.

$$\frac{da(t)}{dt} = -a(t)^{\mathsf{T}} \frac{\partial f(u(t), t, \theta)}{\partial u}, \quad a(t_1) = \frac{\partial \mathcal{L}}{\partial u(t_1)},$$

that dictates the gradient with respect to the parameters. The loss $\mathcal{L}(u(t_1))$ can then be calculated by solving another ODE (the adjoint) rather than backpropagating through the calculations involved in the numerical integration.

However, assuming that the hidden state is governed by an ODE imposes a limitation on the expressiveness of the mapping. For example, Dupont et al. (2019) describes a notable limitation of NODEs is in the inability to represent dynamical systems with intersecting trajectories. In response to such limitations, many works have tried to increase the expressiveness of the mapping. Dupont et al. (2019) proposed to solve the intersection trajectories problem by augmenting the vector space, lifting the points into additional dimensions; Zhu et al. (2021) included time delay in the equation to represent dynamical systems of greater complexity; Massaroli et al. (2021) proposed to condition the vector field on the inputs, allowing the integration limits to be conditioned on the input; Massaroli et al. (2021) and Norcliffe et al. (2020) additionally proposed and proved a second-order ODE system can efficiently solve the intersecting trajectories problem. We note however that the interpretation of NODE as a continuous form of ResNet is also problematic, owing to the fact that the empirical behavior of the ResNet does not match the theoretical properties (Krishnapriyan et al., 2022; Ott et al., 2021). As such, alternative interpretations of the process represented by ODE have been considered. In Zhang et al. (2019), the authors considered an augmentation where the augmented

state corresponds to the parameters of the network governing the latent state. Queiruga et al. (2021) describes the latent evolution through a series of basis functions thereby allowing important concepts such as BatchNorm to be effectively translated in the continuous setting, achieving state-of-the-art performance on a variety of image classification tasks. Further improvements to performance have been made by considering different numerical integrators (Matsubara et al., 2021; Zhuang et al., 2020; 2021).

On a related front, multiple works have attempted to expand NODE to other types of differential equation beyond ODEs. Sun et al. (2019) employed a dictionary method and expanded NODEs to a PDE case, achieving high accuracies both in approximating PDEs and in classifying real-world image datasets. However, Sun et al. (2019) suggested that the method is unstable when training with the adjoint method and therefore is unable to make use of the benefits that come with training with adjoint. Zhang et al. (2018) proposed a density transform approach based on the Monge-Ampere PDE, but did not consider using adjoint-based training. Multiple works have expanded to the stochastic differential equations setting and developed efficient optimization methods for them including (Güler et al., 2019; Jia & Benson, 2019; Kidger et al., 2021a;b; Li et al., 2020; Liu et al., 2019; Xu et al., 2022). Kidger et al. (2020); Morrill et al. (2021a;b) used ideas from rough path theory and controlled differential equations to propose a NODE architecture as a continuous recurrent neural network framework. Salvi et al. (2022) considered stochastic PDEs for spatio-temporal dynamics prediction. Additionally, Chen et al. (2020) models spatio-temporal point-processes using NODEs, and Rubanova et al. (2019); De Brouwer et al. (2019) makes predictions on time series data using NODEs.

## 3 METHOD

We describe the proposed C-NODE method in this section by first providing a brief introduction to the method of characteristics (MoC) for solving PDEs with an illustrative example. We then discuss the types of PDEs we can describe using this method. We finally discuss how we apply the MoC to our C-NODE framework.

### 3.1 METHOD OF CHARACTERISTICS

The MoC provides a procedure for transforming certain PDEs into ODEs along paths known as *characteristics*. In the most general sense, the method applies to general hyperbolic differential equations. We will introduce MoC using a canonical example involving the inviscid Burgers equation, and defer to Griffiths et al. (2015, Chapter 9) for a more complete introduction to the topic. Let $u(x, t) : \mathbb{R} \times \mathbb{R}_+ \to \mathbb{R}$ satisfy the following inviscid Burgers equation

$$\frac{\partial u}{\partial t} + u \frac{\partial u}{\partial x} = 0, \tag{1}$$

where we drop the dependence on $x$ and $t$ for ease of notation. We are interested in the solution of $u$ over some bounded domain $\Omega \subset \mathbb{R} \times \mathbb{R}_+$. Consider parametric forms for the spatial component $x(s) : [0, T] \to \mathbb{R}$ and temporal components $t(s) : [0, T] \to \mathbb{R}_+$ over the fictitious variable $s \in [0, T]$. Intuitively, this allows us to solve an equation on curves $x, t$ that are parameterized by a variable $s$ which we denote $(x(s), t(s))$ as the *characteristic*. Expanding and writing $\mathrm{d}$ as the total derivative, we get

$$\frac{\mathrm{d}}{\mathrm{d}s} u(x(s), t(s)) = \frac{\partial u}{\partial x} \frac{dx}{ds} + \frac{\partial u}{\partial t} \frac{dt}{ds}. \tag{2}$$

Recall the original PDE in equation 1 and substituting the proper terms into equation 2 for $dx/ds = u$, $dt/ds = 1$, $\mathrm{d}u/\mathrm{d}s = 0$, we then recover equation 1. Note that we now have a system of 3 ODEs which we can solve to obtain the characteristics as $x(s) = us + x_0$ and $t(s) = s + t_0$, which are functions of initial conditions $x_0, t_0$. Finally, by solving over a grid of initial conditions $\{x_0^{(i)}\}_{i=1}^{\infty} \in \partial\Omega$, we can obtain the solution of the PDE over $\Omega$. Putting it all together, we have a new ODE that is written as

$$\frac{\mathrm{d}}{\mathrm{d}s} u(x(s), t(s)) = \frac{\partial u}{\partial t} + u \frac{\partial u}{\partial x} = 0,$$

where we can integrate over $s$ through

$$u(x(T), t(T); x_0, t_0) := \int_0^T \frac{\mathrm{d}}{\mathrm{d}s} u(x(s), t(s)) \mathrm{d}s$$

$$:= \int_0^T \frac{\mathrm{d}}{\mathrm{d}s} u(us + x_0, s) \mathrm{d}s,$$

using the adjoint method with boundary conditions $x_0, t_0$. This contrasts the usual direct integration over the variable $t$ that is done in NODE; we now jointly couple the integration through the characteristics. An example of solving this equation over multiple initial conditions is given in Figure 1 with the contrast to standard NODE integration.

**Hyperbolic PDEs in Machine Learning** To motivate using MoC for machine learning problems such as classification or density estimation, we again note that MoC most generally applies to hyperbolic PDEs. These PDEs roughly describe the propagation of physical quantities through time. Such equations may be appropriate for deep learning tasks due to their ability to transport data into different regions of the state space. For instance, in a classification task, we consider the problem of transporting high-dimensional data points that are not linearly separable to spaces where they are linearly separable. Similarly, in generative modeling, we transport a base distribution to data distribution.

## 3.2 NEURAL REPRESENTATION OF CHARACTERISTICS

In the proposed method, we learn the components involved in the MoC, namely the characteristics and the function coefficients. We now generalize the example given in 3.1, which involved two variables, to a $k$-dimensional system. Specifically, consider the following nonhomogeneous boundary value problem (BVP):

$$\begin{cases} \frac{\partial \mathbf{u}}{\partial t} + \sum_{i=1}^{k} a_i(x_1, ..., x_k, \mathbf{u}) \frac{\partial \mathbf{u}}{\partial x_i} = \mathbf{c}(x_1, ..., x_k, \mathbf{u}), & \text{on } \mathbf{x}, t \in \mathbb{R}^k \times [0, \infty) \\ \mathbf{u}(\mathbf{x}(0), 0) = \mathbf{u}_0, & \text{on } \mathbf{x} \in \mathbb{R}^k. \end{cases} \quad (3)$$

Here, $\mathbf{u} : \mathbb{R}^k \times \mathbb{R} \to \mathbb{R}^n$ is a multivariate map, $a_i : \mathbb{R}^{k+n} \to \mathbb{R}$ and $\mathbf{c} : \mathbb{R}^{k+n} \to \mathbb{R}^n$ are functions dependent on values of $\mathbf{u}$ and $x$'s. This problem is well-defined and has a solution as long as $\sum_{i=1}^{k} a_i \frac{\partial \mathbf{u}}{\partial x_i}$ is continuous (Evans, 2010).

MoC is generally used in a scalar context, but the correspondence to the vector case is relatively straightforward. A proof of this can be found in Appendix C.1. To begin, we decompose the PDE in equation 3 into the following system of ODEs

$$\frac{dx_i}{ds} = a_i(x_1, ..., x_k, \mathbf{u}), \quad (4)$$

$$\frac{dt}{ds} = 1, \quad (5)$$

$$\frac{d\mathbf{u}}{ds} = \sum_{i=1}^{k} \frac{\partial \mathbf{u}}{\partial x_i} \frac{dx_i}{ds} = \mathbf{c}(x_1, ..., x_k, \mathbf{u}). \quad (6)$$

We represent this ODE system by parameterizing $dx_i/ds$ and $\partial \mathbf{u}/\partial x_i$ with neural networks. Consequently, $d\mathbf{u}/ds$ is evolving according to equation 6.

Following this expansion, we arrive at

$$\mathbf{u}(\mathbf{x}(T), T) = \mathbf{u}(\mathbf{x}(0), 0) + \int_0^T \frac{d\mathbf{u}}{ds}(\mathbf{x}, \mathbf{u}) \, \mathrm{d}s \quad (7)$$

$$= \mathbf{u}(\mathbf{x}(0), 0) + \int_0^T [\mathbf{J_x u}](\mathbf{x}, \mathbf{u}; \Theta_2) \frac{d\mathbf{x}}{ds}(\mathbf{x}, \mathbf{u}; \Theta_2) \, \mathrm{d}s,$$

where we remove $\mathbf{u}$'s dependency on $\mathbf{x}(s)$ and $\mathbf{x}$'s dependency on $s$ for simplicity of notation. In equation 7, the functions $\mathbf{J_x u}$ and $d\mathbf{x}/ds$ are represented as neural networks with inputs $\mathbf{x}$, $\mathbf{u}$ and parameters $\Theta_2$.

**Conditioning on data** Previous works primarily modeled the task of classifying a set of data points with a fixed differential equation, neglecting possible structural variations lying in the data. Here, we condition C-NODE on each data point, resulting in solving a PDE with a different initial condition and the hyperbolic PDE interpretation of the latent variables. Consider the term given by the integrand in equation 7. The neural network representing the characteristic $d\mathbf{x}/ds$ is conditioned on the input data $\mathbf{z} \in \mathbb{R}^w$. Define a mapping $\mathbf{g}(\cdot) : \mathbb{R}^w \to \mathbb{R}^n$ and we have

$$\frac{dx_i}{ds} = a_i(x_1, \ldots, x_k, \mathbf{u}; \mathbf{g}(\mathbf{z})). \tag{8}$$

By introducing $\mathbf{g}(\mathbf{z})$ in equation 8, the equation describing the characteristics changes depending on the current data point. This leads to the classification task being modeled with a family rather than one single differential equation and allows the C-NODE system to model dynamical systems with intersecting trajectories. This property becomes helpful in Proposition 4.1 in proving that C-NODE can represent intersecting dynamics.

### 3.3 TRAINING C-NODEs

Having introduced the main components of C-NODEs, we can now integrate them into a unified algorithm. To motivate this section, and to be consistent with part of the empirical evaluation, we will consider classification tasks with data $\{(\mathbf{z}_j, \mathbf{y}_j)\}_{j=1}^N$, $\mathbf{z}_j \in \mathbb{R}^w$, $\mathbf{y}_j \in \mathbb{Z}^+$. For instance, $\mathbf{z}_j$ may be an image, and $\mathbf{y}_j$ is its class label. In the approach we pursue here, the image $\mathbf{z}_j$ is first passed through a feature extractor function $\mathbf{g}(\cdot; \Theta_1) : \mathbb{R}^w \to \mathbb{R}^n$ with parameters $\Theta_1$. The output of $\mathbf{g}$ is the feature $\mathbf{u}_0^{(j)} = \mathbf{g}(\mathbf{z}_j; \Theta_1)$ that provides the boundary condition for the PDE on $\mathbf{u}^{(j)}$. We integrate along different characteristic curves indexed by $s \in [0, T]$ with boundary condition $\mathbf{u}^{(j)}(\mathbf{x}(0), 0) = \mathbf{u}_0^{(j)}$, and compute the terminal values as given by equation 7, where we mentioned in Section 3.2,

$$\mathbf{u}^{(j)}(\mathbf{x}(T), T) = \mathbf{u}_0^{(j)} + \int_0^T \mathbf{J}_\mathbf{x}\mathbf{u}^{(i)}\left(\mathbf{x}, \mathbf{u}^{(j)}; \Theta_2\right) \frac{d\mathbf{x}}{ds}\left(\mathbf{x}, \mathbf{u}^{(j)}; \mathbf{u}_0^{(j)}; \Theta_2\right) \mathrm{d}s \tag{9}$$

Finally, $\mathbf{u}^{(j)}(\mathbf{x}(T), T)$ is passed through another neural network, $\Phi(\mathbf{u}^{(j)}(\mathbf{x}(T)); \Theta_3)$ with input $\mathbf{u}^{(j)}(\mathbf{x}(T), T)$ and parameters $\Theta_3$ whose output are the probabilities of each class labels for image $\mathbf{z}_j$. The entire learning process is now reduced to finding optimal weights $(\Theta_1, \Theta_2, \Theta_3)$ which can be achieved by minimizing the loss

$$\mathcal{L} = \sum_{j=1}^N L(\Phi(\mathbf{u}^{(j)}(\mathbf{x}(T), T); \Theta_3), \mathbf{y}_j),$$

where $L(\cdot)$ is the corresponding loss function (e.g. cross entropy in classification). In Algorithm 2, we illustrate the implementation procedure with the forward Euler method for simplicity for the framework but note any ODE solver can be used.

**Combining MoC with Existing NODE Modifications** As mentioned in Section 2, the proposed C-NODEs method can be used as an extension to existing NODE frameworks. In all NODE modifications, the underlying expression of $\int_a^b \mathbf{f}(t, \mathbf{u}; \Theta)\mathrm{d}t$ remains the same. Modifying this expression to $\int_a^b \mathbf{J}_\mathbf{x}\mathbf{u}(\mathbf{x}, \mathbf{u}; \Theta)d\mathbf{x}/ds(\mathbf{x}, \mathbf{u}; \mathbf{u}_0; \Theta)\mathrm{d}s$ results in the proposed C-NODE architecture, with the size of $\mathbf{x}$ being a hyperparameter.

## 4 PROPERTIES OF C-NODEs

C-NODE has a number of theoretical properties that contribute to its expressiveness. We provide some theoretical results on these properties in the proceeding sections. We also define continuous normalizing flows (CNFs) with C-NODEs, extending the CNFs originally defined with NODEs.

**Intersecting trajectories** As mentioned in Dupont et al. (2019), one limitation of NODE is that the mappings cannot represent intersecting dynamics. We prove by construction that conditioning on initial conditions allows C-NODEs to represent some dynamical systems with intersecting trajectories in the following proposition:

**Proposition 4.1.** *The C-NODE can represent a dynamical system on $u(s)$, $du/ds = \mathcal{G}(s, u)$ : $\mathbb{R}_+ \times \mathbb{R} \to \mathbb{R}$, where when $u(0) = 1$, then $u(1) = u(0) + \int_0^1 \mathcal{G}(s, u)ds = 0$; and when $u(0) = 0$, then $u(1) = u(0) + \int_0^1 \mathcal{G}(s, u)ds = 1$.*

*Proof.* See Appendix C.2. $\qquad\square$

**Density estimation with C-NODEs** C-NODEs can also be used to define a continuous density flow that models the density of a variable over space subject to the variable satisfying a PDE, extending the continuous normalizing flows defined with NODEs. For NODEs, if $u(t) \in \mathbb{R}^n$ follows the ODE $du(t)/dt = f(u(t))$, where $f(u(t)) \in \mathbb{R}^n$, then its log likelihood from Chen et al. (2019b, Appendix A) is given by:

$$\frac{\partial \log p(u(t))}{\partial t} = - \operatorname{tr}\left( \frac{df}{du(t)} \right). \tag{10}$$

Similar to the change of log probability of NODEs, as in equation 10, we provide the following proposition for C-NODEs:

**Proposition 4.2.** *Let $u(s)$ be a finite continuous random variable with probability density function $p(u(s))$ and let $u(s)$ satisfy $\frac{du(s)}{ds} = \sum_{i=1}^k \frac{\partial u}{\partial x_i}\frac{dx_i}{ds}$. Assuming $\frac{\partial u}{\partial x_i}$ and $\frac{dx_i}{ds}$ are uniformly Lipschitz continuous in $u$ and continuous in $s$, then the evolution of the log probability of $u$ follows:*

$$\frac{\partial \log p(u(s))}{\partial s} = -\operatorname{tr}\left( \frac{\partial}{\partial u} \sum_{i=1}^k \frac{\partial u}{\partial x_i}\frac{dx_i}{ds} \right)$$

*Proof.* See Appendix C.3. $\qquad\square$

CNFs are continuous and invertible one-to-one mappings onto themselves, i.e., homeomorphisms. Zhang et al. (2020) proved that vanilla NODEs are not universal estimators of homeomorphisms, but augmented neural ODEs (ANODEs) are universal estimators of homeomorphisms. We demonstrate that C-NODEs are pointwise estimators of homeomorphisms, which we formalize in the following proposition:

**Proposition 4.3.** *Given any homeomorphism $h : \Upsilon \to \Upsilon$, $\Upsilon \subset \mathbb{R}^p$, initial condition $u_0$, and time $T > 0$, there exists a flow $u(s, u_0) \in \mathbb{R}^n$ following $\frac{du}{ds} = \frac{\partial u}{\partial x}\frac{dx}{ds} + \frac{\partial u}{\partial t}\frac{dt}{ds}$ such that $u(T, u_0) = h(u_0)$.*

*Proof.* See Appendix C.4. $\qquad\square$

## 5 EXPERIMENTS

We present experiments on image classification tasks, time series prediction tasks, image generation tasks on benchmark datasets, and a synthetic PDE regression task.

### 5.1 CLASSIFICATION EXPERIMENTS WITH IMAGE DATASETS

We first conduct experiments for classification tasks on high-dimensional image datasets, including MNIST, CIFAR-10, and SVHN. We provide results for C-NODE and also combine the framework with existing methods, including ANODEs (Dupont et al., 2019), Input Layer NODEs (IL-NODEs) (Massaroli et al., 2021), and 2nd-Order NODEs (Massaroli et al., 2021). For all classification experiments, we set the encoder of input images for conditioning to be identity, i.e., $g(z) = z$, making the input into C-NODE the original image. This way we focus exclusively on the performance of C-NODE.

The results for the experiments using the adjoint method are reported in Table 1. We investigate the performances of the models on classification accuracy and the number of function evaluations (NFE) taken in the adaptive numerical integration. NFE is an indicator of the model's computational complexity and can be interpreted as the network depth for the continuous NODE (Chen et al., 2019b). Using a similar number of parameters, combining C-NODEs with different models consistently results

in higher accuracy and mostly uses a smaller numbers of NFEs, indicating a better parameter efficiency. An ablation study on C-NODEs' and NODEs' parameters can be found in Appendix E.2. While the average number of function evaluations tends to be lower for C-NODE, we additionally note that, compared to ANODE, training C-NODE with the adjoint method can sometimes have decreased stability. We define instability as having a NFE $> 1000$. To get a rough idea of the differences in stability, when training NODE, C-NODE, and ANODE for image classification on the SVHN dataset for forty instances, NODE appeared unstable six times, C-NODE was unstable three times, and ANODE was never unstable. We note that this was only apparent in the SVHN experiment and when considering C-NODE by itself; the average NFE decreases when adding C-NODE to A-NODE and this was never experienced in the ANODE+C-NODE experiments.

| Dataset | Method | Accuracy $\uparrow$ | NFE $\downarrow$ | Param.[K] $\downarrow$ |
|---------|--------|---------------------|-----------------|------------------------|
| SVHN | NODE | $75.28 \pm 0.836\%$ | 131 | 115.444 |
| | C-NODE | $\mathbf{82.19 \pm 0.478}\%$ | **124** | 113.851 |
| | ANODE | $89.8 \pm 0.952\%$ | 167 | 112.234 |
| | ANODE+C-NODE | $\mathbf{92.23 \pm 0.176}\%$ | **146** | 112.276 |
| | 2nd-Ord | $88.22 \pm 1.11\%$ | 161 | 112.801 |
| | 2nd-Ord+C-NODE | $\mathbf{92.37 \pm 0.118}\%$ | **135** | 112.843 |
| | IL-NODE | $89.69 \pm 0.369\%$ | 195 | 113.368 |
| | IL-NODE+C-NODE | $\mathbf{93.31 \pm 0.088}\%$ | **95** | 113.752 |
| CIFAR-10 | NODE | $56.30 \pm 0.742\%$ | 152 | 115.444 |
| | C-NODE | $\mathbf{64.28 \pm 0.243}\%$ | **151** | 113.851 |
| | ANODE | $70.99 \pm 0.483\%$ | **177** | 112.234 |
| | ANODE+C-NODE | $\mathbf{71.36 \pm 0.220}\%$ | 224 | 112.276 |
| | 2nd-Ord | $70.84 \pm 0.360\%$ | 189 | 112.801 |
| | 2nd-Ord+C-NODE | $\mathbf{73.68 \pm 0.153}\%$ | **131** | 112.843 |
| | IL-NODE | $72.55 \pm 0.238\%$ | 134 | 113.368 |
| | IL-NODE+C-NODE | $\mathbf{73.78 \pm 0.154}\%$ | **85** | 113.752 |
| MNIST | NODE | $96.90 \pm 0.154\%$ | 72 | 85.468 |
| | C-NODE | $\mathbf{97.56 \pm 0.431}\%$ | 72 | 83.041 |
| | ANODE | $99.12 \pm 0.021\%$ | 68 | 89.408 |
| | ANODE+C-NODE | $\mathbf{99.20 \pm 0.002}\%$ | **60** | 88.321 |
| | 2nd-Ord | $99.35 \pm 0.002\%$ | **52** | 89.552 |
| | 2nd-Ord+C-NODE | $\mathbf{99.38 \pm 0.037}\%$ | 61 | 88.465 |
| | IL-NODE | $99.33 \pm 0.039\%$ | **53** | 89.597 |
| | IL-NODE+C-NODE | $\mathbf{99.33 \pm 0.001}\%$ | 60 | 88.51 |

Table 1: Mean test results over 5 runs of different NODE models over SVHN, CIFAR-10, and MNIST. Accuracy and NFE at convergence are reported. Applying C-NODE always increases models' accuracy and usually reduces models' NFE as well as the standard error.

## 5.2 TIME SERIES PREDICTION WITH C-NODES

We test C-NODEs, NODEs, and augmented versions of C-NODEs and NODEs on the time series prediction problem using 100 sequences of each of the following datasets: the PhysioNet dataset, containing measurements from the first 48 hours of patients' admissions to ICU (Goldberger et al., 2000 (June 13); the Hopper dataset, containing physical simulation results generated with the Hopper model from the Deepmind Control Suite (Rubanova et al., 2019; Tassa et al., 2018); the Human Activity dataset in the UCI dataset, containing time series data on twelve features from five individuals walking, sitting, and lying (Dua & Graff, 2017); and the synthetic ODE dataset, generated with a first-order ODE whose initial condition follows a Gaussian distribution, and Gaussian noises are added to the observations (Rubanova et al., 2019). The PhysioNet and Human Activity datasets both contain stochastic processes while the Hopper dataset and the ODE dataset are deterministic. Since NODE, ANODE, and C-NODE model deterministic processes, the Hopper dataset and the ODE

| Dataset | Method | MSE $\downarrow$ | Param.[K] $\downarrow$ |
|---|---|---|---|
| PHYSIONET | NODE | $(1.17 \pm 0.027) \times 10^{-2}$ | 76.663 |
| | C-NODE | $(\mathbf{1.07 \pm 0.026}) \times \mathbf{10^{-2}}$ | 76.205 |
| | ANODE | $(1.15 \pm 0.036) \times 10^{-2}$ | 86.127 |
| | C-NODE + ANODE | $(\mathbf{1.04 \pm 0.027}) \times \mathbf{10^{-2}}$ | 85.705 |
| HUMAN ACTIVITY | NODE | $(1.998 \pm 0.62) \times 10^{-1}$ | 51.042 |
| | C-NODE | $(\mathbf{1.797 \pm 0.59}) \times \mathbf{10^{-1}}$ | 50.964 |
| | ANODE | $(1.140 \pm 0.048) \times 10^{-1}$ | 69.948 |
| | C-NODE + ANODE | $(\mathbf{9.52 \pm 0.67}) \times \mathbf{10^{-2}}$ | 69.397 |
| HOPPER | NODE | $(5.87 \pm 0.23) \times 10^{-2}$ | 44.776 |
| | C-NODE | $(\mathbf{5.68 \pm 0.11}) \times \mathbf{10^{-2}}$ | 44.218 |
| | ANODE | $(5.68 \pm 0.10) \times 10^{-2}$ | 51.976 |
| | C-NODE + ANODE | $(\mathbf{4.96 \pm 0.16}) \times \mathbf{10^{-2}}$ | 51.506 |
| ODE | NODE | $(4.73 \pm 0.17) \times 10^{-2}$ | 43.231 |
| | C-NODE | $(\mathbf{4.17 \pm 1.34}) \times \mathbf{10^{-2}}$ | 42.935 |
| | ANODE | $(4.60 \pm 0.53) \times 10^{-2}$ | 43.767 |
| | C-NODE + ANODE | $(\mathbf{4.01 \pm 1.23}) \times \mathbf{10^{-2}}$ | 43.556 |

Table 2: Mean test results over 4 runs of different NODE models over PhysioNet, Human Activity, Hopper, and ODE. Test mean square errors are reported. Applying C-NODE always reduces the test error, and mostly reduces' the standard error. Training dynamics are shown in Figures 3, 4, 5, 6 in Appendix A.2.

dataset readily fit the modeling frameworks. On the other hand, the PhysioNet and Human Activity datasets require augmenting the dynamics of the NODE models with stochasticity to model the arrival of events.

We follow the experimental setup for interpolation tasks in Rubanova et al. (2019), where we define an autoregressive model with the encoder being an ODE-RNN model and the decoder being a latent differential equation[2]. The main purpose of this experiment is to compare the ODE-RNN when using C-NODE versus NODE. ODE-RNN is a standard method for including ODE modeling in time series tasks as described in Chen et al. (2019b) and Rubanova et al. (2019). We consider interpolation tasks by first encoding the time series $\{x_i, t_i\}_{i=0}^{N}$ of length $N$ and computing the approximate posterior $q(z_0 | \{x_i, t_i\}_{i=0}^{N}) = \mathcal{N}(\mu_{z_0}, \sigma_{z_0})$ as done in Rubanova et al. (2019). Then, $\mu(z_0), , \sigma(z_0)$ are computed as $\varrho(\text{ODE-RNN}_\phi(\{x_i, t_i\}_{i=0}^{N}))$, where $\varrho$ is a function that encodes the terminal hidden states into mean and variance of the latent variable $z_0$. ODE-RNN$(\cdot)$ is a model whose states obey an ODE between observations and are updated according to new observations as described in Rubanova et al. (2019). To predict the state at an observation time $t_i$, we sample initial states $z_0$ from the posterior which are then decoded using another neural network. We finally average generated observations at each observation time to compute the test data errors. The results are presented in Table 2 and suggest that C-NODE based models use slightly fewer parameters while achieving lower error rates than NODE models. C-NODE models the latent dynamics as a first order PDE, which is a natural extension of the ODE model that NODE uses.

## 5.3 CONTINUOUS NORMALIZING FLOW WITH C-NODES

We compare the performance of CNFs defined with NODEs to flows defined with C-NODEs on MNIST and CIFAR-10. We use the Hutchinson trace estimator to calculate the trace and use multi-scale convolutional architectures to model the density transformation as done in (Dinh et al., 2017; Grathwohl et al., 2019) [1]. Differential equations are solved using the Runge-Kutta method of order 5 of the Dormand-Prince-Shampine solver and trained with the adjoint method. Although the Euler

---

[2]This is based on the code of Rubanova et al. (2019) provided at `https://github.com/YuliaRubanova/latent_ode`

[1]This is based on the code that the authors of (Grathwohl et al., 2019) provided in `https://github.com/rtqichen/ffjord`

| Model | MNIST | | | CIFAR-10 | | |
|---|---|---|---|---|---|---|
| | B/D ↓ | Param. ↓ | NFE ↓ | B/D ↓ | Param. ↓ | NFE ↓ |
| Real NVP (Dinh et al., 2017) | 1.05 | N/A | – | 3.49 | N/A | – |
| Glow (Kingma & Dhariwal, 2018) | 1.06 | N/A | – | 3.35 | 44.0M | – |
| RQ-NSF (Durkan et al., 2019) | – | – | – | 3.38 | 25.2M | – |
| Res. Flow (Chen et al., 2019a) | 0.97 | 16.6M | – | **3.28** | 25.2M | – |
| CP-Flow (Huang et al., 2021) | 1.02 | 2.9M | – | 3.40 | 1.9M | – |
| NODE | 1.00 | **335.1K** | 1350 | 3.49 | 410.1K | 1847 |
| C-NODE | **0.95** | 338.0K | **1323** | 3.44 | **406.0K** | **1538** |

Table 3: Experimental results on generation tasks, with NODE, C-NODE, and other models. B/D indicates Bits/dim. Using a similar number of parameters, C-NODE outperforms NODE on all three datasets, and has a significantly lower NFE when training for CIFAR-10.

forward method is faster, experimental results show that its fixed step size often leads to negative Bits/Dim, indicating the importance of adaptive solvers. As shown in table 3 and figure 7, using a similar number of parameters, experimental results show that CNFs defined with C-NODEs perform better than CNFs defined with NODEs in terms of Bits/Dim, as well as having lower variance, and using a lower NFE on both MNIST and CIFAR-10.

## 5.4 PDE MODELING WITH C-NODES

We consider a regression example for a hyperbolic PDE with a known analytical solution. Since NODEs assume that the latent state is only dependent on a scalar (namely time), they cannot model dependencies that vary over multiple spatial variables required by most PDEs. We modify the assumptions used in the classification and density estimation experiments where the boundary conditions were constant as in equation 3. We approximate the following BVP:

$$\begin{cases} u\frac{\partial u}{\partial x} + \frac{\partial u}{\partial t} = u, \\ u(x,0) = 2t, \qquad 1 \le x \le 2 \end{cases} \tag{11}$$

which has an analytical solution given by $u(x,t) = \frac{2x\exp(t)}{2\exp(t)+1}$. We generate a training dataset by randomly sampling 200 points $(x,t)$, $x \in [1,2]$, $t \in [0,1]$, as well as values $u(x,t)$ at those points. We test C-NODE and NODE on 200 points randomly sampled as $(x,t) \in [1,2] \times [0,1]$. For this experiment, C-NODE uses 809 parameters while NODE uses 1185 parameters. We quantify the differences in the representation capabilities by examining how well each method can represent the PDE. C-NODE deviates 8.05% from the test set, while NODE deviates 30.52%. Further experimental details can be found in Appendix A.4.1.

## 6 DISCUSSION

We describe an approach for extending NODEs to the case of PDEs by solving a series of ODEs along the characteristics of a PDE. The approach applies to any black-box ODE solver and can combine with existing NODE-based frameworks. We empirically showcase its efficacy on classification tasks while demonstrating its success in improving convergence using Euler forward method without the adjoint method. C-NODEs also consistently achieve lower testing MSEs over different time series prediction datasets, while having lower standard errors. Additionally, C-NODEs empirically achieve better performances on density estimation tasks, while being more efficient with the number of parameters and using lower NFEs. C-NODE's efficiency over physical modeling is also highlighted with additional experiments. Discussion on limitations can be found in Appendix B.

## ACKNOWLEDGMENTS

This work was supported in part by the Office of Naval Research (ONR) under grant number N00014-21-1-2590. AH was supported by NSF-GRFP.

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

## A  EXPERIMENTAL DETAILS

Implementation details of this paper can be found at `https://github.com/XingziXu/NeuralPDE.git`.

### A.1  EXPERIMENTAL DETAILS OF CLASSIFICATION TASKS

We report the average performance over five independent training processes, and the models are trained for 100 epochs for all three datasets. We also report the training dynamics of C-NODE and NODE using the adjoint sensitivity method and the euler backpropagation, as shown in Figure 2.

As shown in Figure 2, using the Euler solver, it appears that C-NODEs converge faster than the vanilla NODEs (usually in one epoch) while generally having a more stable training process with smaller variance. Additionally, on experiments with MNIST, C-NODEs converge in only one epoch, while NODEs converge in roughly 15 epochs. This provides additional empirical evidence on the benefits of training using the characteristics.

The input for 2nd-Ord, NODE, and C-NODE are the original images. In the IL-NODE, we transform the input to a latent space before the integration by the integral; that is, we raise the $\mathbb{R}^{c \times h \times w}$ dimensional input image into the $\mathbb{R}^{(c+p) \times h \times w}$ dimensional latent feature space[3].

For SVHN and CIFAR-10, we assume $\mathbf{x} \in \mathbb{R}^3$, i.e., the Jacobian $\mathbf{J_x u} = (\partial\mathbf{u}/\partial x_1, \partial\mathbf{u}/\partial x_2, \partial\mathbf{u}/\partial x_3)$. We model each partial derivative $\partial\mathbf{u}/\partial x_i$ with a separate convolutional network. The network architecture for the network modeling the partial derivatives is as shown in Table 6. The network architecture for the network modeling $d\mathbf{x}/ds$ is as shown in Table 6. The architecture in Tables 5, 6 are used for both CIFAR-10 and SVHN. Note that the network architecture for MNIST differs slightly due to the lower dimensionality of MNIST. The hyperparameters used are as shown in Table 4.

---

[3]This is based on the code of Massaroli et al. (2021) provide in `https://github.com/DiffEqML/torchdyn`

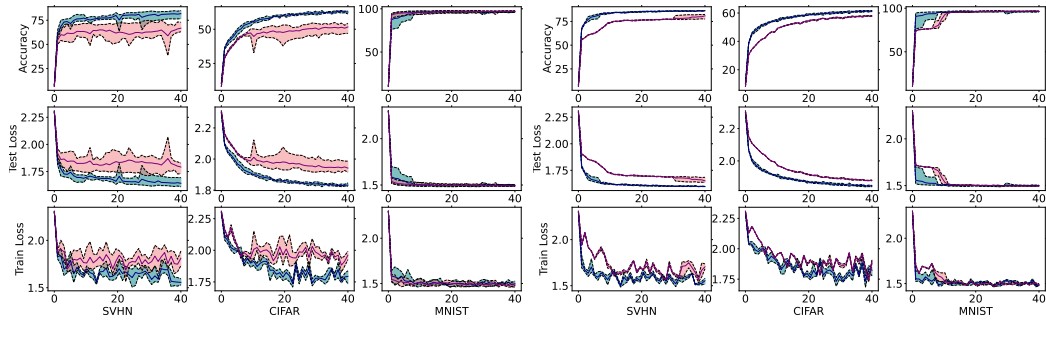

(a) Adjoint training;                                    (b) Backprop through Euler training;

Figure 2: **Red: NODE. Blue: C-NODE.** Training dynamics of different datasets with adjoint in Fig. 2a and with Euler in Fig. 2b averaged over five runs. The first column is the training process of SVHN, the second column is of CIFAR-10, and the third column is of MNIST. By incorporating the C-NODE method, we achieve a more stable training process in both CIFAR-10 and SVHN, while achieving higher accuracy. Full-sized figure in supplementary materials.

We decode the result after performing the continuous transformations along characteristics curves, back to the $\mathbb{R}^{c \times h \times w}$ dimensional object space. Combining this with the C-NODE can be seen as solving a PDE on the latest features of the images rather than on the images directly.

Unlike ODEs, we take derivatives with respect to different variables in PDEs. For a PDE with $k$ variables, this results in the constraint of the balance equations

$$\frac{\partial^2 u}{\partial x_i x_j} = \frac{\partial^2 u}{\partial x_j x_i}, \ i, j \in \{1, 2, ..., k\}, i \neq j.$$

This can be satisfied by defining the $k$-th derivative with a neural network, and integrate $k - 1$ times to get the first order derivatives. Another way of satisfying the balance equation is to drop the dependency on the variables, i.e., $\forall i \in \{1, 2, ..., k\}$,

$$\frac{\partial u}{\partial x_i} = f_i(u; \theta).$$

When we drop the dependency, all higher order derivatives are zero, and the balance equations are satisfied.

All experiments were performed on NVIDIA RTX 3090 GPUs on a cloud cluster.

| Data | MNIST | SVHN | CIFAR10 |
|---|---|---|---|
| Model | CNN | CNN | CNN |
| # of PDE Dimensions | 2 | 3 | 3 |
| Optimizer | AdamW | AdamW | AdamW |
| Learning Rate | 1.00E-3 | 1.00E-3 | 1.00E-3 |
| Weight Decay | 5.00E-04 | 5.00E-04 | 5.00E-04 |

Table 4: Training hyperparameters for image classification.

## A.2   EXPERIMENTAL RESULTS AND DETAILS OF TIME SERIES PREDICTIONS

### A.2.1   EXPERIMENTAL DETAILS OF TIME SERIES PREDICTIONS ON REAL-WORLD DATASETS

We provide a more detailed explanation of the time-series experiments which are based on the ODE-RNN framework described in Rubanova et al. (2019). Our experiments test the effectiveness of NODE, C-NODE, ANODE, and their combinations under the ODE-RNN framework by computing

| Operation Layer | Kernel Size | Stride | Padding | Dilation | Size of Output |
|---|---|---|---|---|---|
| Input | N/A | N/A | N/A | N/A | $3 \times 32 \times 32$ |
| Convolution Layer | 3 | 1 | 0 | 1 | $60 \times 32 \times 32$ |
| ReLU | N/A | N/A | N/A | N/A | $60 \times 32 \times 32$ |
| Convolution Layer | 3 | 1 | 0 | 1 | $60 \times 32 \times 32$ |
| ReLU | N/A | N/A | N/A | N/A | $60 \times 32 \times 32$ |
| Convolution Layer | 3 | 1 | 0 | 1 | $3 \times 32 \times 32$ |

Table 5: Network architecture of the network modeling terms in the Jacobian.

| Operation Layer | Kernel Size | Stride | Padding | Dilation | Size of Output |
|---|---|---|---|---|---|
| Input | N/A | N/A | N/A | N/A | $4 \times 32 \times 32$ |
| Convolution Layer | 1 | 1 | 0 | 1 | $8 \times 32 \times 32$ |
| ReLU | N/A | N/A | N/A | N/A | $8 \times 32 \times 32$ |
| Convolution Layer | 3 | 1 | 1 | 0 | $8 \times 32 \times 32$ |
| ReLU | N/A | N/A | N/A | N/A | $8 \times 32 \times 32$ |
| Convolution Layer | 1 | 1 | 0 | 1 | $3 \times 32 \times 32$ |
| Flatten | N/A | N/A | N/A | N/A | $3072 \times 1$ |
| Linear | N/A | N/A | N/A | N/A | $3 \times 1$ |
| ReLU | N/A | N/A | N/A | N/A | $3 \times 1$ |

Table 6: Network architecture of the network modeling the characteristic curve $d\mathbf{x}/ds$.

evaluation metrics under the different ODE models. The ODE-RNN framework involves combining the strengths of both a neural ODE and RNN to initially embed the time series history into a latent distribution parameterized by an RNN and then decode the predicted latent distribution into the original data space. The main components of the method are illustrated in Algorithm 1. Forecasts are computed by integrating the latent space in time according to the neural ODE model with initial condition $z_0$ distributed according to the parameterization by the RNN. We define the approximate posterior of $z_0$ as $q(z_0|\{x_i, t_i\}_{i=0}^N) = \mathcal{N}(\mu_{z_0}, \sigma_{z_0})$, where $\mu_{z_0}, \sigma_{z_0} = \varrho(\text{ODE-RNN}_\phi(\{x_i, t_i\}_{i=0}^N))$. To bypass the RNN requirement of a fixed observation rate, the ODE-RNN model uses states that obey an ODE in between observations and are updated at new observations. Given a set of time series data $\{x_i, t_i\}_{i=0}^N$, we embed the time series using the ODE-RNN. We then pass the output of the ODE-RNN model through a function $\varrho(\cdot)$ to get the mean $\mu_{z_0}$ and the variance $\sigma_{z_0}$ of the posterior of $z_0$. To predict the value at timestamp $T$, we sample $K$ initial conditions $z_0$ from the posterior $q(z_0|\{x_i, t_i\}_{i=0}^N)$, integrate the latent ODE model until timestamp $T$ and use the average of the $K$ integrations as the result.

For all experiments, we follow the experimental setup as described in `https://github.com/YuliaRubanova/latent_ode`. We experiment with the ODE-RNN framework using NODE, C-NODE, ANODE, and their combinations as the ODE backbone of both the ODE-RNN model and the latent ODE. Training NODE follows the original setup, with the dimension of the ODE model in the ODE-RNN being 20, the number of units per layer in each of GRU update networks being 100, the number of units per layer in ODE function being around 100, the number of layers in ODE function in generative and the recognition ODE both being 1.

We use a C-NODE with a dimensionality of 8. The network architecture details are given in Tables 7, 8. The training hyperparameters are given in Table 9. The number of units per layer in the network describing $d\mathbf{x}/d\mathbf{s}$ is 12. For the network describing $\partial\mathbf{u}/\partial x_i$, the dimension of the ODE model in the ODE-RNN is 20, the number of units per layer in each of GRU update networks is 100, the number of units per layer in the ODE function is tuned to match the number of parameters in the NODE models, the number of layers in ODE function in generative and the recognition ODE is 1.

---

**Algorithm 1** Algorithm for ODE-RNN model(Rubanova et al., 2019)

**Input:** Data points and their timestamps $\{(x_i, t_i)\}_{i=1,...,N}$
$h_0 = \mathbf{0}$
**for** $i$ in 1, 2, ..., N **do**
$\quad h_i' = \text{ODESolve}(f_\theta, h_{i-1}, (t_{i-1}, t_i))$
$\quad h_i = \text{RNNCell}(h_i', x_i)$
**end for**
$o_i = \text{OutputNN}(h_i)$ for all $i = 1, ..., N$
**Return:** $\{o_i\}_{i=1,...,N}; h_N$

---

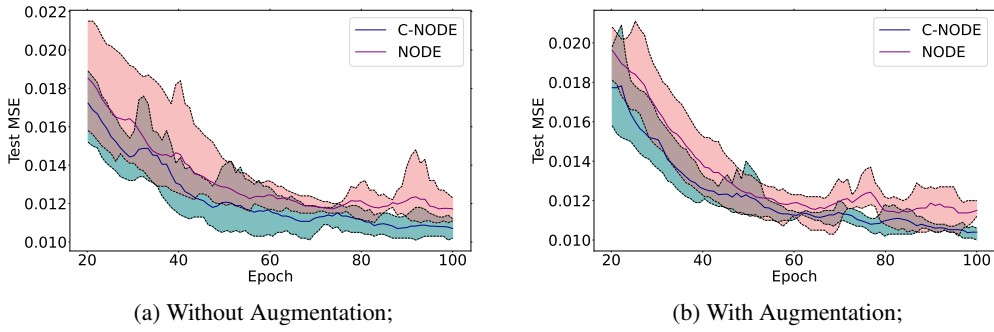

(a) Without Augmentation;                    (b) With Augmentation;

Figure 3: **Red: NODE. Blue: C-NODE.** Training dynamics of CNODE, NODE, and their augmented versions on Physionet dataset. C-NODE achieves lower testing MSE, and has a more stable training dynamics

| Operation Layer | Input Features | Output Features |
|---|---|---|
| Linear Layer | 20 | 70 |
| Tanh | N/A | N/A |
| Linear Layer | 70 | 70 |
| Tanh | N/A | N/A |
| Linear Layer | 70 | 160 |

Table 7: Network Structure of $\partial\mathbf{u}/\partial\mathbf{x}_i$ when using ODE dataset

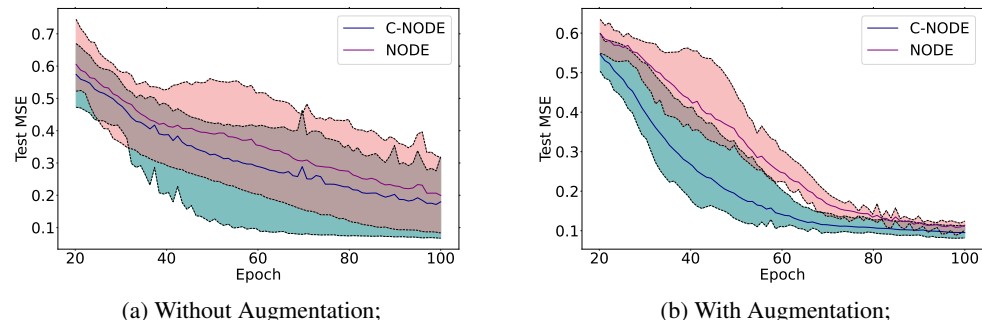

(a) Without Augmentation;                    (b) With Augmentation;

Figure 4: **Red: NODE. Blue: C-NODE.** Training dynamics of CNODE, NODE, and their augmented versions on Human Activity dataset. C-NODE achieves lower testing MSE, and has a more stable training dynamics

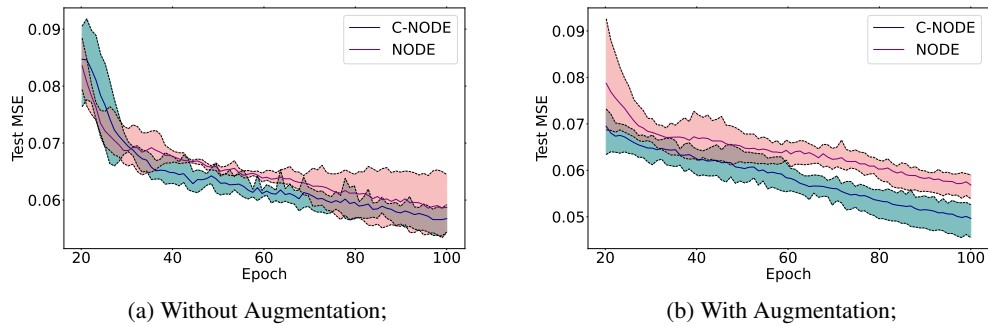

(a) Without Augmentation;

(b) With Augmentation;

Figure 5: **Red: NODE. Blue: C-NODE.** Training dynamics of CNODE, NODE, and their augmented versions on Hopper dataset. C-NODE achieves lower testing MSE, and has a more stable training dynamics

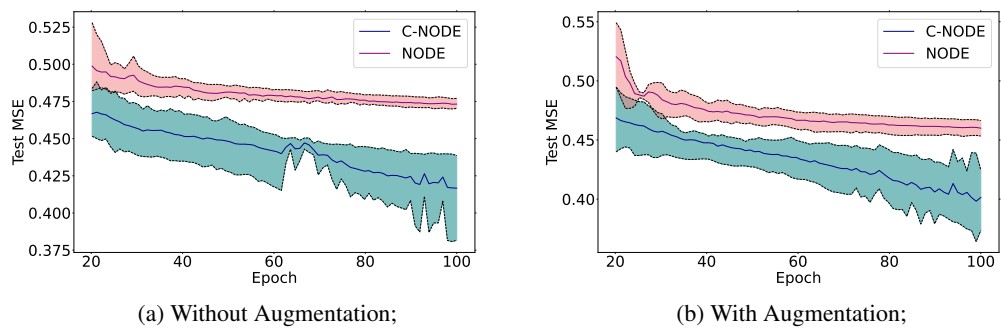

(a) Without Augmentation;

(b) With Augmentation;

Figure 6: **Red: NODE. Blue: C-NODE.** Training dynamics of CNODE, NODE, and their augmented versions on a toy ODE dataset. C-NODE achieves a significantly lower testing MSE

| Operation Layer | Input Features | Output Features |
|---|---|---|
| Linear Layer | 2 | 12 |
| Tanh | N/A | N/A |
| Linear Layer | 12 | 12 |
| Tanh | N/A | N/A |
| Linear Layer | 12 | 8 |

Table 8: Network Structure of $d\mathbf{x}/ds$ when using ODE dataset

| Data | Physionet | Activity | Hopper | Periodic |
|---|---|---|---|---|
| # of PDE Dimensions | 8 | 8 | 8 | 7 |
| Optimizer | Adamax | Adamax | Adamax | Adamax |
| Learning Rate | 1.0E-2 | 1.0E-2 | 1.0E-2 | 1.0E-2 |
| Weight Decay | 0.0 | 0.0 | 0.0 | 0.0 |

Table 9: Training hyperparameters for time series analysis

## A.2.2 EXPERIMENT RESULTS OF TIME SERIES PREDICTIONS ON SYNTHETIC DATASET

We test C-NODEs, ANODEs, and NODEs on a synthetic time series prediction problem. We define a function by $u(x,t) = \frac{2x \exp(t)}{2 \exp(t)+1}$, and we sample $\tilde{u} = u(x,t) + 0.1\epsilon_t$, where $\epsilon_t \sim \mathcal{N}(0,1)$ over $x \in [1,2]$, $t \in [0,1]$ to generate the training dataset. We test the performance on $t \in [n, n+1]$ with $n \in \{0, 1, \ldots, 5\}$.

| Time | [0,1] | [1,2] | [2,3] | [3,4] | [4,5] | [5,6] |
|------|-------|-------|-------|-------|-------|-------|
| NODE | 0.0322 | 0.1764 | 0.4681 | 0.8093 | 1.1911 | 1.6202 |
| ANODE | 0.0428 | 0.0629 | 0.1248 | 0.2778 | 0.5360 | 0.9252 |
| C-NODE | **0.0270** | **0.0365** | **0.0582** | **0.1474** | **0.3300** | **0.6054** |

Table 10: Time series prediction results for NODE, ANODE, and C-NODE at different time intervals. Errors are testing mean squared errors. Across all time intervals, C-NODE outperforms NODE and ANODE.

We also test C-NODEs, NODEs, and ANODEs on time series prediction with different levels of noise. Specifically, using the same function as above, we form training and testing dataset with $\epsilon_t \sim \mathcal{N}(0,m)$, $m \in \{0, 1, \ldots, 5\}$. We test the performance on the time period $t \in [0,1]$.

| Noise Level | 0 | 1 | 2 | 3 | 4 | 5 |
|-------------|---|---|---|---|---|---|
| NODE | 0.0326 | 0.1784 | 0.7886 | 1.9685 | 3.7530 | 6.1553 |
| ANODE | 0.04 | 0.1984 | 0.6035 | 1.0574 | 1.4850 | **2.0593** |
| C-NODE | **0.0267** | **0.1011** | **0.3294** | **0.7148** | **1.2856** | 2.0834 |

Table 11: Time series prediction results for NODE, ANODE, and C-NODE at different noise levels. Errors are testing mean squared errors.

## A.2.3 EXPERIMENTAL DETAILS OF TIME SERIES PREDICTIONS ON SYNTHETIC DATASET

We consider the task of predicting $u(x,t) = \frac{2 \cdot x \cdot e^t}{2 \cdot e^t + 1}$ at different times $t$, over $x \in [1,2]$. We specify the initial condition of $u(1,0)$.

We use a 8 dimensional C-NODE network. The result is calculated with

$$u(x,t) = u(1,0) + \int_0^t \sum_{i=1}^8 \frac{\partial u}{\partial z_i} \frac{dz_i}{ds} ds.$$

NODE is calculated with

$$u(x,t) = u(1,0) + \int_0^t \frac{\partial u}{\partial t} dt.$$

The experiment results are given in table 11.

In our experiments, C-NODEs use 1221 parameters, ANODEs use 1270 parameters, NODEs use 1290 parameters.

All experiments were performed on NVIDIA RTX 3080 ti GPUs on a local machine.

## A.3 EXPERIMENTAL DETAILS OF CONTINUOUS NORMALIZING FLOWS

We report the average performance over four independent training processes. As shown in Figure 8, compared to NODE, using a C-NODE structure improves the stability of training, as well as having better performance. Specifically, the standard errors for C-NODEs on MNIST, SVHN, and CIFAR-10 are 0.37%, 0.51%, and 0.24% respectively, and for NODEs the standard errors on MNIST, SVHN, and CIFAR-10 are 1.07%, 0.32%, and 0.22% respectively. The training dynamics of CNFs on MNIST

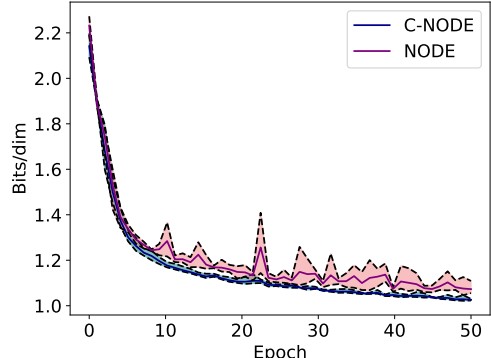

Figure 7: **Red: NODE. Blue: C-NODE.** Training dynamics of CNFs on MNIST dataset with adjoint method. We present Bits/dim of the first 50 training epochs.

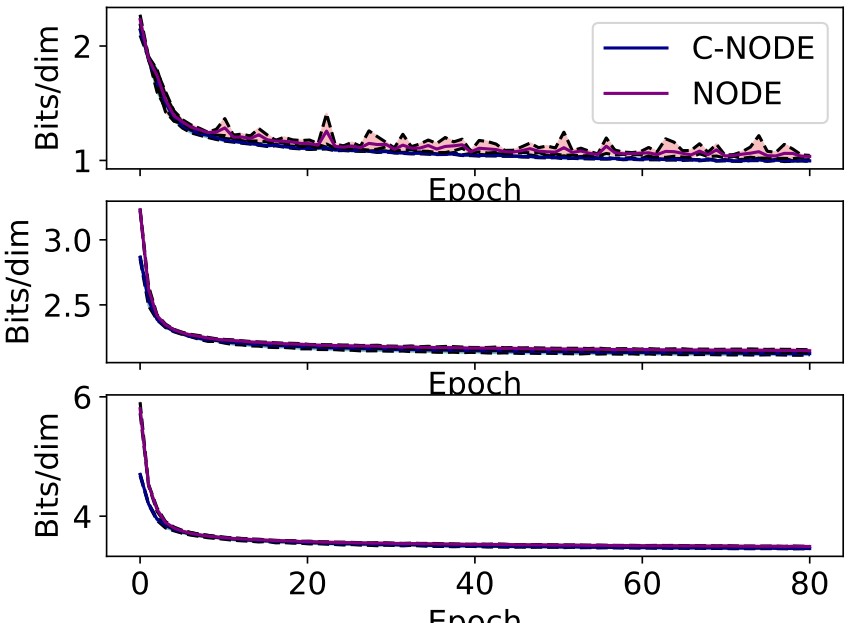

Figure 8: The training process averaged over 4 runs of C-NODE and NODE. The first row are the results on MNIST, the second row are the results on SVHN, the third row are the results on CIFAR-10.

are reported in Figure 7. C-NODE consistently achieves lower Bits/dim across the training process, while having more stable training dynamics.

The network structures are provided in Tables 12, 13. The training hyperparameters are provided in Table 14. The experiments are developed using code adapted from the code that the authors of Grathwohl et al. (2019) provided in `https://github.com/rtqichen/ffjord`.

All experiments were performed on NVIDIA RTX 3090 GPUs on a cloud cluster.

| Operation Layer | Input Channels | Output Channels | Kernel Size | Stride | Padding |
|---|---|---|---|---|---|
| Convolutional Layer | 4 | 8 | $(3,3)$ | $(1,1)$ | $(1,1)$ |
| Tanh | N/A | N/A | N/A | N/A | N/A |
| Convulutional Layer | 9 | 70 | $(3,3)$ | $(1,1)$ | $(1,1)$ |
| Tanh | N/A | N/A | N/A | N/A | N/A |
| Convolutional Layer | 71 | 70 | $(3,3)$ | $(1,1)$ | $(1,1)$ |
| Tanh | N/A | N/A | N/A | N/A | N/A |
| Convolutional Layer | 71 | 8 | $(3,3)$ | $(1,1)$ | $(1,1)$ |
| Tanh | N/A | N/A | N/A | N/A | N/A |
| Convolutional Layer | 9 | 3 | $(3,3)$ | $(1,1)$ | $(1,1)$ |

Table 12: Network structure of $\partial \mathbf{u}/\partial \mathbf{x}_i$ when using CIFAR-10 dataset

| Oper. Layer | Input Chan. | Output Chan. | Kern. | Stride | Pad. | Input Fea. | Output Fea. |
|---|---|---|---|---|---|---|---|
| Conv. Layer | 4 | 12 | (3,3) | (1,1) | (1,1) | N/A | N/A |
| SiLU | N/A | N/A | N/A | N/A | N/A | N/A | N/A |
| Conv. Layer | 12 | 12 | (3,3) | (1,1) | (1,1) | N/A | N/A |
| SiLU | N/A | N/A | N/A | N/A | N/A | N/A | N/A |
| Conv. Layer | 12 | 3 | (3,3) | (1,1) | (1,1) | N/A | N/A |
| Flatten | N/A | N/A | N/A | N/A | N/A | N/A | N/A |
| Linear Layer | 71 | 8 | (3,3) | (1,1) | (1,1) | 3072 | 2 |

Table 13: Network structure of $d\mathbf{x}/ds$ when using CIFAR-10 dataset

| Data | MNIST | SVHN | CIFAR-10 |
|---|---|---|---|
| # of PDE Dimensions | 2 | 3 | 3 |
| Optimizer | Adam | Adam | Adam |
| Learning Rate | 1.00E-3 | 1.00E-3 | 1.00E-3 |
| Weight Decay | 0.0 | 0.0 | 0.0 |

Table 14: Training hyperparameters for continuous normalizing flow

## A.4 EXPERIMENTAL DETAILS OF PDE MODELING

### A.4.1 2-DIMENSIONAL BURGER'S EQUATION

We want to solve the initial value problem

$$\begin{cases} u\frac{\partial u}{\partial x} + \frac{\partial u}{\partial t} = u, \\ u(x,0) = 2x, \quad 1 \le x \le 2, \end{cases}$$

where the exact solution is $u(x,t) = \frac{2xe^t}{(2e^t+1)}$. Our dataset's input are 200 randomly sampled points $(x,t)$, $x \in [1,2]$, $t \in [0,1]$, and the dataset's outputare the exact solutions at those points.

For the C-NODE architecture, we define four networks: $NN_1(x,t)$ for $\frac{\partial u}{\partial x}$, $NN_2(x,t)$ for $\frac{\partial u}{\partial t}$, $NN_3(t)$ for the characteristic path $(x(s),t(s))$, $NN_4(x)$ for the initial condition. The result is calculated in four steps:

1. Integrate $\Delta u = \int_0^t \frac{du(x(s),t(s))}{ds}ds = \int_0^t \frac{\partial u}{\partial t}\frac{dt}{ds} + \frac{\partial u}{\partial x}\frac{dx}{ds}ds = NN_2 * NN_3[0] + NN_1 * NN_3[1]ds$ as before.
2. Given $x$, $t$, solve equation $\iota + NN_3(NN_4(\iota))[0] * t = x$ for $\iota$ iteratively, with $\iota_{n+1} = x - NN_3(NN_4(\iota_n))[0] * t$. $\iota_0$ is initialized to be $x$.
3. Calculate initial value $u(x(0),t(0)) = NN_4(\iota)$.
4. $u(x,t) = \Delta u + u(x(0),t(0))$.

For the NODE architecture, we define one network: $NN_1(x,t)$ for $\frac{\partial u}{\partial t}$. The result is calculated as $u(x,t) = \int_0^t \frac{\partial u}{\partial t}dt = \int_0^t NN_1 dt$.

All experiments were performed on NVIDIA RTX 3080-TI GPUs on a local machine.

### A.4.2 100-DIMENSIONAL CONVECTION EQUATION

We additionally experiment on solving a 100-dimensional convection equation given by:

$$\begin{cases} \frac{\partial u}{\partial t} = -\mathrm{div}(\mu'(t)u(\mathbf{x},t)) \\ u(\mathbf{x},0) = \exp(-\frac{1}{2}\|\mathbf{x} - \mu(0)\|^2) \\ \mu(t) = \sin\left(\begin{bmatrix} 0 \\ 0.01t \\ \vdots \\ t \end{bmatrix}\right) \end{cases}$$

where $\mathbf{x} \in \mathbb{R}^{100}$, $u : \mathbb{R}^{100} \times \mathbb{R}_+ \to \mathbb{R}$, $\mu : \mathbb{R}_+ \to \mathbb{R}^{100}$.

This equation has an analytical solution $u(\mathbf{x},t) = \exp(-\frac{1}{2}\|\mathbf{x} - \mu(t)\|^2)$.

We generate a training dataset with 1000 points, and a testing dataset with 1000 points. We uniformly sample $\mathbf{x}$, with each $x_i \in [0, 0.5]$ and $t \in [0, 10]$. We calculate the output using the analytical solution.

We define three networks for C-NODE. The first network models $d\mathbf{x}/ds : \mathbb{R}^{101} \to \mathbb{R}^{101}$, the second network models $\mathbf{J}_{\mathbf{x}}u : \mathbb{R}^{101} \to \mathbb{R}^{101}$, the third network $\Gamma : \mathbb{R}^{101} \to \mathbb{R}$ models the initial condition $u(\mathbf{x},0)$. We first integrate $(\mathbf{x},t)$ using the first two networks. The output of these networks is then input to the third network to obtain the value of the solution. The total number of parameters for the networks is 11611.

We define two networks for NODE. The first network models $d\mathbf{u}/ds : \mathbb{R}^{101} \to \mathbb{R}^{101}$, the second network $\Gamma : \mathbb{R}^{101} \to \mathbb{R}$ models the initial condition $u(\mathbf{x},0)$. We first integrate $(\mathbf{x},t)$ using the first network. The output is $\mathbb{R}^{101}$. We put the output into the second network, and arrive at the output. The number of parameters used here is 14214.

We test C-NODE and NODE on the dataset with Gaussian noises at different magnitudes. As shown in Table 15, C-NODE performs 14.2% better than NODE when there is no noise, 44.4% better

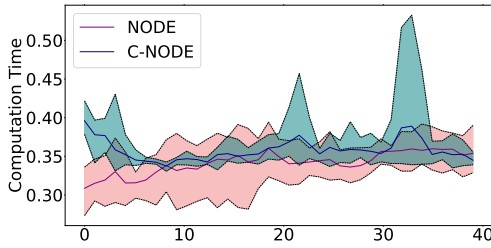 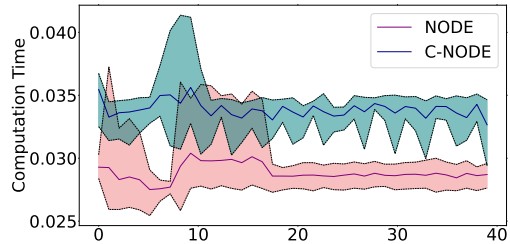

(a) Computation Time With Adjoint Method;  (b) Computation Time With Euler Forward Method;

Figure 9: **Red: NODE. Blue: C-NODE.** Computation time at each epochs using the adjoint method and Euler forward integration. C-NODE uses slightly more computation time for Euler forward integration, and the same amount of time when using the adjoint method.

when there's 10% noise, 39.18% better when there's 20% noise, and 40.1% better when there's 30% noise. The architecture of C-NODE and NODE are as given in Tables 16, 17, 19, 18. The models are optimized with the AdamW optimizer, with a learning rate of $5 \times 10^{-3}$ and a weight decay of $5 \times 10^{-4}$.

We also report computation time of NODE and C-NODE using the adjoint method and the Euler forward integration. As shown in Figure 9, C-NODE uses slightly more computation time for Euler forward integration, and the same amount of time when using the adjoint method. Although each call of C-NODE uses slightly more time, NODE uses a bigger number of functional evaluations. This results in roughly a similar computation time for NODE and C-NODE when integrated using the adjoint method.

**A note on the difference between physics-informed neural networks (PINNs)**    PINNs (Raissi et al., 2019) is a method that solves a PDE using a regularization approach by minimizing a regularization term enforcing the PDE. Comparing this to C-NODE, there are a few apparent differences. First, PINNs is a very general framework that can accommodate almost any type of PDE. This contrasts with C-NODE which only computes solutions to hyperbolic PDEs. On the other hand, PINNs generally scale poorly with dimension due to the difficulty in computing high dimensional derivatives. Besides, PINNs require the exact form of the PDEs being solved, whereas C-NODE does not require this information.

| Noise Level | 0 | 10% | 20% | 30% |
|---|---|---|---|---|
| NODE | 0.0500 | 0.0644 | 0.0675 | 0.0688 |
| C-NODE | 0.0438 | 0.0446 | 0.0485 | 0.0491 |

Table 15: C-NODE performs better than NODE on all noise levels. C-NODE uses 11611 parameters, NODE uses 14215 parameters.

| Operation Layer | Input Channels | Output Channels |
|---|---|---|
| Linear Layer | 101 | 16 |
| Softplus | N/A | N/A |
| Linear Layer | 16 | 16 |
| Softplus | N/A | N/A |
| Linear Layer | 16 | 101 |

Table 16: Network structure of $\mathbf{J_x u}$

| Operation Layer | Input Channels | Output Channels |
|---|---|---|
| Linear Layer | 102 | 16 |
| Softplus | N/A | N/A |
| Linear Layer | 16 | 16 |
| Softplus | N/A | N/A |
| Linear Layer | 16 | 101 |

Table 17: Network structure of $d\mathbf{x}/ds$

| Operation Layer | Input Channels | Output Channels |
|---|---|---|
| Linear Layer | 101 | 32 |
| ReLU | N/A | N/A |
| Linear Layer | 32 | 32 |
| ReLU | N/A | N/A |
| Linear Layer | 32 | 1 |

Table 18: Network structure of Regressor

| Operation Layer | Input Channels | Output Channels |
|---|---|---|
| Linear Layer | 101 | 40 |
| Softplus | N/A | N/A |
| Linear Layer | 40 | 40 |
| Softplus | N/A | N/A |
| Linear Layer | 40 | 101 |

Table 19: Network structure of NODE

## B  LIMITATIONS

There are several limitations to the proposed method. The MoC only applies to hyperbolic PDEs, and we only consider first-order semi-linear PDEs in this paper. This may be a limitation since this is a specific class of PDEs that does not model all data. We also did not enforce any particular structure to prevent characteristics from intersecting, which may result in shock waves and rarefactions. However, we believe that this is unlikely to happen due to the high dimensionality of the ambient space. As noted in the experiments section, C-NODE can have decreased stability compared to ANODE when defined by exceeding an extreme threshold of number of function evaluations.

## C  APPROXIMATION CAPABILITIES OF C-NODE

**Proposition C.1** (Method of Characteristics for Vector Valued PDEs). *Let* $\mathbf{u}(x_1, \ldots, x_k) : \mathbb{R}^k \to \mathbb{R}^n$ *be the solution of a first order semilinear PDE on a bounded domain* $\Omega \subset \mathbb{R}^k$ *of the form*

$$\sum_{i=1}^{k} a_i(x_1, \ldots, x_k, \mathbf{u}) \frac{\partial \mathbf{u}}{\partial x_i} = \mathbf{c}(x_1, \ldots, x_k, \mathbf{u}) \quad on \ (x_1, \ldots, x_k) = \mathbf{x} \in \Omega. \tag{12}$$

*Additionally, let* $\mathbf{a} = (a_1, \ldots, a_k)^T : \mathbb{R}^{k+n} \to \mathbb{R}^k, \mathbf{c} : \mathbb{R}^{k+n} \to \mathbb{R}^n$ *be Lipschitz continuous functions. Define a system of ODEs as*

$$\begin{cases} \frac{d\mathbf{x}}{ds}(s) &= \mathbf{a}(\mathbf{x}(s), \mathbf{U}(s)) \\ \frac{d\mathbf{U}}{ds}(s) &= \mathbf{c}(\mathbf{x}(s), \mathbf{U}(s)) \\ \mathbf{x}(0) &:= \mathbf{x}_0, \ \mathbf{x}_0 \in \partial\Omega \\ \mathbf{u}(\mathbf{x}_0) &:= \mathbf{u}_0 \\ \mathbf{U}(0) &:= \mathbf{u}_0 \end{cases}$$

*where* $\mathbf{x}_0$ *and* $\mathbf{u}_0$ *define the initial condition,* $\partial\Omega$ *is the boundary of the domain* $\Omega$*. Given initial conditions* $\mathbf{x}_0, \mathbf{u}_0$*, the solution of this system of ODEs* $\mathbf{U}(s) : [a, b] \to \mathbb{R}^d$ *is equal to the solution of the PDE in Equation equation 12 along the characteristic curve defined by* $\mathbf{x}(s)$*, i.e.,* $\mathbf{u}(\mathbf{x}(s)) = \mathbf{U}(s)$*. The union of solutions* $\mathbf{U}(s)$ *for all* $\mathbf{x}_0 \in \partial\Omega$ *is equal to the solution of the original PDE in Equation equation 12 for all* $\mathbf{x} \in \Omega$*.*

**Lemma C.2** (Gronwall's Lemma (Howard, 1998)). *Let* $U \subset \mathbb{R}^n$ *be an open set. Let* $\mathbf{f} : U \times [0, T] \to \mathbb{R}^n$ *be a continuous function and let* $\mathbf{h_1}, \mathbf{h_2} : [0, T] \to U$ *satisfy the initial value problems:*

$$\frac{d\mathbf{h_1}(t)}{dt} = f(\mathbf{h_1}(t), t), \ \mathbf{h_1}(0) = \mathbf{x_1},$$

$$\frac{d\mathbf{h_2}(t)}{dt} = f(\mathbf{h_2}(t), t), \ \mathbf{h_2}(0) = \mathbf{x_2}.$$

*If there exists non-negative constant* $C$ *such that for all* $t \in [0, T]$

$$\|\mathbf{f}(\mathbf{h_2}(t), t) - \mathbf{f}(\mathbf{h_1}(t), t)\| \le C \|\mathbf{h_2}(t) - \mathbf{h_1}(t)\|,$$

*where* $\| \cdot \|$ *is the Euclidean norm. Then, for all* $t \in [0, T]$*,*

$$\|\mathbf{h_2}(t) - \mathbf{h_1}(t)\| \le e^{Ct} \|\mathbf{x_2} - \mathbf{x_1}\|.$$

### C.1  PROOF OF PROPOSITION C.1

This proof is largely based on the proof for the univarate case provided at[4]. We extend for the vector valued case.

*Proof.* For PDE on $\mathbf{u}$ with $k$ input, and an $n$-dimensional output, we have $a_i : \mathbb{R}^{k+n} \to \mathbb{R}, \frac{\partial \mathbf{u}}{\partial x_i} \in \mathbb{R}^n$, and $\mathbf{c} : \mathbb{R}^{k+n} \to \mathbb{R}^n$. In proposition C.1, we look at PDEs in the following form

$$\sum_{i=1}^{k} a_i(x_1, \ldots, x_k, \mathbf{u}) \frac{\partial \mathbf{u}}{\partial x_i} = \mathbf{c}(x_1, \ldots, x_k, \mathbf{u}). \tag{13}$$

---

[4]https://en.wikipedia.org/wiki/Method_of_characteristics#Proof_for_quasilinear_Case

Defining and substituting $\mathbf{x} = (x_1, \ldots, x_k)^\intercal$, $\mathbf{a} = (a_1, \ldots, a_k)^\intercal$, and Jacobian $\mathbf{J}(\mathbf{u}(\mathbf{x})) = (\frac{\partial \mathbf{u}}{\partial x_1}, \ldots, \frac{\partial \mathbf{u}}{\partial x_k}) \in \mathbb{R}^{n \times k}$ into Equation equation 12 result in

$$\mathbf{J}(\mathbf{u}(\mathbf{x}))\mathbf{a}(\mathbf{x}, \mathbf{u}) = \mathbf{c}(\mathbf{x}, \mathbf{u}). \tag{14}$$

From proposition C.1, the characteristic curves are given by

$$\frac{dx_i}{ds} = a_i(x_1, \ldots, x_k, \mathbf{u}),$$

and the ODE system is given by

$$\frac{d\mathbf{x}}{ds}(s) = \mathbf{a}(\mathbf{x}(s), \mathbf{U}(s)), \tag{15}$$

$$\frac{d\mathbf{U}}{ds}(s) = \mathbf{c}(\mathbf{x}(s), \mathbf{U}(s)). \tag{16}$$

Define the difference between the solution to equation 16 and the PDE in equation 12 as

$$\Delta(s) = \|\mathbf{u}(\mathbf{x}(s)) - \mathbf{U}(s)\|^2 = (\mathbf{u}(\mathbf{x}(s)) - \mathbf{U}(s))^\intercal (\mathbf{u}(\mathbf{x}(s)) - \mathbf{U}(s)),$$

Differentiating $\Delta(s)$ with respect to $s$ and plugging in equation 15, we get

$$\begin{aligned}
\Delta'(s) := \frac{d\Delta(s)}{ds} &= 2(\mathbf{u}(\mathbf{x}(s)) - \mathbf{U}(s)) \cdot (\mathbf{J}(\mathbf{u})\mathbf{x}'(s) - \mathbf{U}'(s)) \\
&= 2[\mathbf{u}(\mathbf{x}(s)) - \mathbf{U}(s)] \cdot [\mathbf{J}(\mathbf{u})\mathbf{a}(\mathbf{x}(s), \mathbf{U}(s)) - \mathbf{c}(\mathbf{x}(s), \mathbf{U}(s))]. \tag{17}
\end{aligned}$$

equation 14 gives us $\sum_{i=1}^{k} a_i(x_1, \ldots, x_k, \mathbf{u})\frac{\partial \mathbf{u}}{\partial x_i} - \mathbf{c}(x_1, \ldots, x_k, \mathbf{u}) = 0$. Plugging this equality into equation 17 and rearrange terms, we have

$$\begin{aligned}
\Delta'(s) = 2[\mathbf{u}(\mathbf{x}(s)) - \mathbf{U}(s)] \cdot \{&[\mathbf{J}(\mathbf{u})\mathbf{a}(\mathbf{x}(s), \mathbf{U}(s)) - \mathbf{c}(\mathbf{x}(s), \mathbf{U}(s))] \\
&- [\mathbf{J}(\mathbf{u})\mathbf{a}(\mathbf{x}(s), \mathbf{u}(s)) - \mathbf{c}(\mathbf{x}(s), \mathbf{u}(s))]\}.
\end{aligned}$$

Combining terms, we have

$$\begin{aligned}
\Delta' &= 2(\mathbf{u} - \mathbf{U}) \cdot ([\mathbf{J}(\mathbf{u})\mathbf{a}(\mathbf{U}) - \mathbf{c}(\mathbf{U})] - [\mathbf{J}(\mathbf{u})\mathbf{a}(\mathbf{u}) - \mathbf{c}(\mathbf{u})]) \\
&= 2(\mathbf{u} - \mathbf{U}) \cdot (\mathbf{J}(\mathbf{u})[\mathbf{a}(\mathbf{U}) - \mathbf{a}(\mathbf{u})] + [\mathbf{c}(\mathbf{U}) - \mathbf{c}(\mathbf{u})]).
\end{aligned}$$

Applying triangle inequality, we have

$$\|\Delta'\| \leq 2\|\mathbf{u} - \mathbf{U}\|(\|\mathbf{J}(\mathbf{u})\|\|\mathbf{a}(\mathbf{U}) - \mathbf{a}(\mathbf{u})\| + \|\mathbf{c}(\mathbf{U}) - \mathbf{c}(\mathbf{u})\|).$$

By the assumption in proposition C.1, $\mathbf{a}$ and $\mathbf{c}$ are Lipschitz continuous. By Lipschitz continuity, we have $\|\mathbf{a}(\mathbf{U}) - \mathbf{a}(\mathbf{u}))\| \leq A\|\mathbf{u} - \mathbf{U}\|$ and $\|\mathbf{c}(\mathbf{U}) - \mathbf{c}(\mathbf{u}))\| \leq B\|\mathbf{u} - \mathbf{U}\|$, for some constants A and B in $\mathbb{R}_+$. Also, for compact set $[0, s_0]$, $s_0 < \infty$, since both $\mathbf{u}$ and Jacobian $\mathbf{J}$ are continuous mapping, $\mathbf{J}(\mathbf{u})$ is also compact. Since a subspace of $\mathbb{R}^n$ is compact if and only it is closed and bounded, $\mathbf{J}(\mathbf{u})$ is bounded (Strichartz, 2000). Thus, $\|\mathbf{J}(\mathbf{u})\| \leq M$ for some constant $M$ in $\mathbb{R}_+$. Define $C = 2(AM + B)$, we have

$$\begin{aligned}
\|\Delta'(s)\| &\leq 2(AM\|\mathbf{u} - \mathbf{U}\| + B\|\mathbf{u} - \mathbf{U}\|)\|\mathbf{u} - \mathbf{U}\| \\
&= C\|\mathbf{u} - \mathbf{U}\|^2 \\
&= C\|\Delta(s)\|.
\end{aligned}$$

From proposition C.1, we have $\mathbf{u}(\mathbf{x}(0)) = \mathbf{U}(0)$. As proved above, we have

$$\left\|\frac{d\mathbf{u}(\mathbf{x}(s))}{ds} - \frac{d\mathbf{U}(s)}{ds}\right\| := \|\Delta'(s)\| \leq C\|\Delta(s)\|,$$

where $C < \infty$. Thus, by lemma C.2, we have

$$\|\Delta(s)\| \leq e^{Ct}\|\Delta(0)\| = e^{Ct}\|\mathbf{u}(\mathbf{x}(0)) - \mathbf{U}(0)\| = 0.$$

This further implies that $\mathbf{U}(s) = \mathbf{u}(\mathbf{x}(s))$, so long as $\mathbf{a}$ and $\mathbf{c}$ are Lipschitz continuous. $\qquad \square$

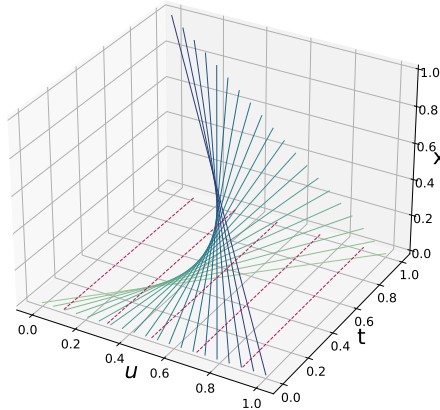

Figure 10: Comparison of C-NODEs and NODEs. C-NODEs (solid blue) learn a family of integration paths conditioned on the input value, avoiding intersecting dynamics. NODEs (dashed red) integrate along a 1D line that is not conditioned on the input value and can not represent functions requiring intersecting dynamics.

## C.2 PROOF OF PROPOSITION 4.1

*Proof.* Suppose have C-NODE given by

$$\frac{du}{ds} = \frac{\partial u}{\partial x}\frac{dx}{ds} + \frac{\partial u}{\partial t}\frac{dt}{ds}.$$

Write out specific functions for these terms to match the desired properties of the function. Define initial condition $u(0,0) = u_0$. By setting

$$\frac{dx}{ds}(s, u_0, \theta) = 1, \qquad\qquad \frac{dt}{ds}(s, u_0, \theta) = u_0,$$

$$\frac{\partial u}{\partial x}(u(x,t), \theta) = 1, \qquad\qquad \frac{\partial u}{\partial t}(u(x,t), \theta) = -2,$$

have the ODE and solution,

$$\frac{du}{ds} = 1 - 2u_0$$

$$\implies u(s; u_0) = (1 - 2u_0)\, s$$

$$\implies u\left(s; \begin{bmatrix} 0 \\ 1 \end{bmatrix}\right) = \left(1 - 2\begin{bmatrix} 0 \\ 1 \end{bmatrix}\right) s = \begin{bmatrix} 1 \\ -1 \end{bmatrix} s.$$

To be specific, we can represent this system with the following family of PDEs:

$$\frac{\partial u}{\partial x} + u_0 \frac{\partial u}{\partial t} = 1 - 2u_0.$$

We can solve this system to obtain a function that has intersecting trajectories. The solution is visualized in Figure 10, which shows that C-NODE can be used to learn and represent this function $\mathcal{G}$. It should be noted that this is not the only possible solution to function $\mathcal{G}$, as when $\partial t/\partial s = 0$, we fall back to a NODE system with the dynamical system conditioned on the input data. In this conditioned setting, we can then represent $\mathcal{G}$ by stopping the dynamics at different times $t$ as in (Massaroli et al., 2021).

$\square$

## C.3 PROOF OF PROPOSITION 4.2

The proof uses the change of variables formula for a particle that depends on a vector rather than a scalar and it follows directly from the proof given in (Chen et al., 2019b, Appendix A). We provide the full proof for completeness.

*Proof.* We initially assume that $\sum_{i=1}^{k} \frac{\partial u}{\partial x_i} \frac{dx_i}{ds}$ is Lipschitz continuous in $u$ and continuous in $t$, so every initial value problem has a unique solution (Evans, 2010). We additionally assume $u(s)$ is bounded.

We want to show that the probability flow satisfies

$$\frac{\partial p(u(s))}{\partial s} = \text{tr} \left( \frac{\partial}{\partial u} \sum_{i=1}^{k} \frac{\partial u}{\partial x_i} \frac{dx_i}{ds} \right).$$

Define $T_\epsilon = u(s + \epsilon)$. The discrete change of variables states that $u_1 = f(u_0) \Rightarrow \log p(u_1) = \log p(u_0) - \log|\det \frac{\partial f}{\partial u_0}|$ (Rezende & Mohamed, 2015).

Taking the limit of the time difference between $u_0$ and $u_1$, and using the definition of derivatives,

$$
\begin{aligned}
\frac{\partial \log p(u(s))}{\partial t} &= \lim_{\epsilon \to 0^+} \frac{\log p(u(s+\epsilon)) - \log p(u(s))}{\epsilon} \\
&= \lim_{\epsilon \to 0^+} \frac{\log p(u(s)) - \log|\det \frac{\partial}{\partial u} T_\epsilon(u(t))| - \log p(u(s))}{\epsilon} \\
&= -\lim_{\epsilon \to 0^+} \frac{\log|\det \frac{\partial}{\partial u} T_\epsilon(u(s))|}{\epsilon} \\
&= -\lim_{\epsilon \to 0^+} \frac{\frac{\partial}{\partial \epsilon} \log|\det \frac{\partial}{\partial u} T_\epsilon(u(s))|}{\frac{\partial}{\partial \epsilon} \epsilon} \\
&= -\lim_{\epsilon \to 0^+} \frac{\partial}{\partial \epsilon} \log|\det \frac{\partial}{\partial u} T_\epsilon(u(s))| - \lim_{\epsilon \to 0^+} \frac{\partial}{\partial \epsilon} \log|\det \frac{\partial}{\partial u} T_\epsilon(u(s))| \\
&= -\lim_{\epsilon \to 0^+} \frac{1}{|\det \frac{\partial}{\partial u} T_\epsilon(u(s))|} \frac{\partial}{\partial \epsilon} |\det \frac{\partial}{\partial u} T_\epsilon(u(s))| \\
&= -\frac{\lim_{\epsilon \to 0^+} \frac{\partial}{\partial \epsilon} |\det \frac{\partial}{\partial u} T_\epsilon(u(s))|}{\lim_{\epsilon \to 0^+} |\det \frac{\partial}{\partial u} T_\epsilon(u(s))|} \\
&= -\lim_{\epsilon \to 0^+} \frac{\partial}{\partial \epsilon} |\det \frac{\partial}{\partial u} T_\epsilon(u(s))|
\end{aligned}
$$

The Jacobi's formula states that if $A$ is a differentiable map from the real numbers to $n \times n$ matrices, then $\frac{d}{dt} \det A(t) = \text{tr} \left( \text{adj}(A(t)) \frac{dA(t)}{dt} \right)$, where $\text{adj}$ denotes the adjugate matrix. Applying this relation, we obtain

$$
\begin{aligned}
\frac{\partial \log p(u(t))}{\partial t} &= -\lim_{\epsilon \to 0^+} \text{tr} \left[ \text{adj} \left( \frac{\partial}{\partial u} T_\epsilon(u(s)) \right) \frac{\partial}{\partial \epsilon} \frac{\partial}{\partial u} T_\epsilon(u(s)) \right] \\
&= -\text{tr} \left[ \left( \lim_{\epsilon \to 0^+} \text{adj} \left( \frac{\partial}{\partial u} T_\epsilon(u(t)) \right) \right) \left( \lim_{\epsilon \to 0^+} \frac{\partial}{\partial \epsilon} \frac{\partial}{\partial u} T_\epsilon(u(s)) \right) \right] \\
&= -\text{tr} \left[ \text{adj} \left( \frac{\partial}{\partial u} u(t) \right) \lim_{\epsilon \to 0^+} \frac{\partial}{\partial \epsilon} \frac{\partial}{\partial u} T_\epsilon(u(s)) \right] \\
&= -\text{tr} \left[ \lim_{\epsilon \to 0^+} \frac{\partial}{\partial \epsilon} \frac{\partial}{\partial u} T_\epsilon(u(s)) \right]
\end{aligned}
$$

Substituting $T_\epsilon$ with its Taylor series expansion and taking the limit, we have the desired result

$$
\begin{aligned}
\frac{\partial \log p(u(t))}{\partial t} &= -\operatorname{tr}\left( \lim_{\epsilon \to 0^+} \frac{\partial}{\partial \epsilon} \frac{\partial}{\partial u} \left( u + \epsilon \frac{du}{ds} + \mathcal{O}(\epsilon^2) + \mathcal{O}(\epsilon^3) + \dots \right) \right) \\
&= -\operatorname{tr}\left( \lim_{\epsilon \to 0^+} \frac{\partial}{\partial \epsilon} \frac{\partial}{\partial u} \left( u + \epsilon \sum_{i=1}^{k} \frac{\partial u}{\partial x_i} \frac{dx_i}{ds} + \mathcal{O}(\epsilon^2) + \mathcal{O}(\epsilon^3) + \dots \right) \right) \\
&= -\operatorname{tr}\left( \lim_{\epsilon \to 0^+} \frac{\partial}{\partial \epsilon} \left( I + \frac{\partial}{\partial u} \epsilon \sum_{i=1}^{k} \frac{\partial u}{\partial x_i} \frac{dx_i}{ds} + \mathcal{O}(\epsilon^2) + \mathcal{O}(\epsilon^3) + \dots \right) \right) \\
&= -\operatorname{tr}\left( \lim_{\epsilon \to 0^+} \left( \frac{\partial}{\partial u} \sum_{i=1}^{k} \frac{\partial u}{\partial x_i} \frac{dx_i}{ds} + \mathcal{O}(\epsilon) + \mathcal{O}(\epsilon^2) + \dots \right) \right) \\
&= -\operatorname{tr}\left( \frac{\partial}{\partial u} \sum_{i=1}^{k} \frac{\partial u}{\partial x_i} \frac{dx_i}{ds} \right)
\end{aligned}
$$

$\square$

### C.4 Proof of Proposition 4.3

*Proof.* To prove proposition 4.3, we need to show that for any homeomorphism $h(\cdot)$, there exists a $u(s, u_0) \in \mathbb{R}^n$ following a C-NODE system such that $u(s = T, u_0) = h(u_0)$.

Without loss of generality, we suppose that $T = 1$.

Defining a C-NODE system as:

$$
\begin{cases}
\frac{du}{ds} = \frac{\partial u}{\partial x} \frac{dx}{ds} + \frac{\partial u}{\partial t} \frac{dt}{ds}, \\
\frac{dx}{ds}(s, u_0) = 1, \\
\frac{\partial u}{\partial x}(u(x,t)) = h(u_0), \\
\frac{dt}{ds}(s, u_0) = u_0, \\
\frac{\partial u}{\partial t}(u(x,t)) = -1.
\end{cases}
$$

Then, $\frac{du}{ds} = h(u_0) - u_0$. At $s = 1$, have

$$
\begin{aligned}
u(s = 1, u_0) &= u(s = 0, u_0) + \int_0^1 \frac{du}{ds} ds \\
&= u_0 + \int_0^1 \frac{\partial u}{\partial x} \frac{dx}{ds} + \frac{\partial u}{\partial t} \frac{dt}{ds} ds \\
&= u_0 + \int_0^1 h(u_0) \cdot 1 + (-1) \cdot u_0 ds \\
&= u_0 + h(u_0) - u_0 \\
&= h(u_0).
\end{aligned}
$$

The inverse map will be defined by integration backwards. Specifically, we have

$$
\begin{aligned}
u(s = 0, u_0) &= u(s = 1, u_0) + \int_1^0 \frac{du}{ds} ds \\
&= h(u_0) - \int_0^1 \frac{\partial u}{\partial x} \frac{dx}{ds} + \frac{\partial u}{\partial t} \frac{dt}{ds} ds \\
&= h(u_0) - \int_0^1 h(u_0) \cdot 1 + (-1) \cdot u_0 ds \\
&= h(u_0) - h(u_0) + u_0 \\
&= u_0.
\end{aligned}
$$

Thus, for any homeomorphism $h(\cdot)$, there exists a C-NODE system, such that forward integration for time $s = 1$ is equivalent as applying $h(\cdot)$, and backward integration for time $s = 1$ is equivalent to applying $h^{-1}(\cdot)$. □

## D  INCLUDING INITIAL STATE IN NODE'S INPUT

Conditioning on the initial condition allows NODE to model intersecting trajectories. We compare NODE conditioned on initial values and C-NODE's performance on image classification tasks on MNIST, SVHN, and CIFAR-10 datasets, using the forward Euler solver and adjoint solver.

As shown in Figure 11, by conditioning on initial values, NODE's performances are improved over all datasets when using adjoint integration, and improved on SVHN dataset when using Euler forward integration. When using Euler forward integration, C-NODE performs better on all three datasets than NODE and conditioned-NODE. C-NODE higher accuracy within fewer epochs and has lower variance throughout the training process. All methods use a comparable numbers of parameters with C-NODE using the fewest as reported in Table 20. When using adjoint method, C-NODE performs better on all datasets than NODE, and performs better on SVHN and CIFAR-10 datasets than conditioned-NODE, while achieving similar performance to conditioned-NODE.

It is worth noting that the characteristic curves of the C-NODE also depend on the initial values, which allows C-NODE to model a dynamical system with intersecting trajectories. C-NODE and conditioned NODE are two different methods of including the initial values. Empirically, C-NODE achieves better results and is more efficient with parameters, as suggested in Figure 11 and Table 20.

| Dataset | Method | Accuracy (Euler) ↑ | Accuracy (Adaptive) ↑ | Param. [K] ↓ |
|---------|--------|--------------------|-----------------------|--------------|
| SVHN | NODE | $82.42 \pm 0.043$ | $64.510 \pm 1.637$ | 115.444 |
| | COND | $83.30 \pm 0.131$ | $80.39 \pm 1.088$ | 116.399 |
| | C-NODE | $86.28 \pm 0.086$ | $82.35 \pm 1.458$ | 113.851 |
| CIFAR-10 | NODE | $59.492 \pm 0.366$ | $52.13 \pm 1.464$ | 115.444 |
| | COND | $56.77 \pm 0.137$ | $57.71 \pm 0.226$ | 118.447 |
| | C-NODE | $61.97 \pm 0.139$ | $63.57 \pm 0.421$ | 113.851 |
| MNIST | NODE | $97.09 \pm 0.024$ | $96.79 \pm 0.374$ | 85.468 |
| | COND | $93.38 \pm 1.798$ | $98.05 \pm 0.112$ | 87.287 |
| | C-NODE | $96.93 \pm 0.103$ | $97.26 \pm 0.276$ | 83.041 |

Table 20: Parameter counts and classification accuracy for different models and integration schemes in Conditional-NODE experiment.

All experiments were performed on NVIDIA RTX 3090 GPUs on a cloud cluster.

## E  ABLATION STUDY

### E.1  ABLATION STUDY ON DIMENSION OF C-NODE

We perform an ablation study on the impact of the number of dimensions of the C-NODE we implement. This study allows us to evaluate the relationship between the model performance and the model's limit of mathematical approximating power. Empirical results show that as we increase the number of dimensions used in the C-NODE model, the C-NODE's performance first improves and then declines, due to overfitting. We have found out that information criteria like AIC and BIC can be successfully applied for dimension selection in this scenario.

In previous experiments, we represent $\partial \mathbf{u}/\partial x_i$ with separate and independent neural networks $\mathbf{c}_i(\mathbf{u}, \theta)$. Here, we represent all $k$ functions as a vector-valued function $[\partial \mathbf{u}/\partial x_1, ..., \partial \mathbf{u}/\partial x_k]^T$. We approximate this vector-valued function with a neural network $\mathbf{c}(\mathbf{u}, \theta)$. The model is trained using the Euler solver to have better training stability when the neural network has a large number of parameters. Experiment details for the ablation study is as shown in Figures 12, 13, 14.

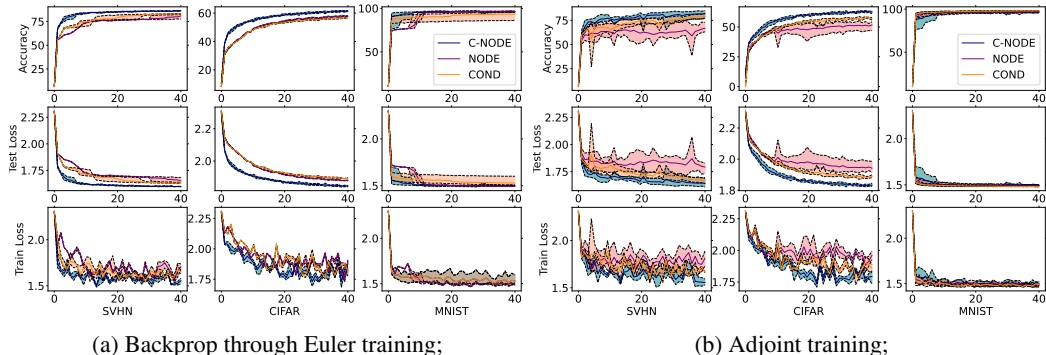

(a) Backprop through Euler training;                    (b) Adjoint training;

Figure 11: **Red: NODE. Blue: C-NODE. Orange: Conditional NODE** Training dynamics of different datasets with adjoint in Fig. 11a and with Euler in Fig. 11b averaged over four runs. The first column is the training process of SVHN, the second column is of CIFAR-10, and the third column is of MNIST. By conditioning on the initial values, NODE's performances are improved on all datasets when using adjoint integration, and improved on SVHN dataset when using the Euler backpropagation.

### E.2    ABLATION STUDY ON NUMBER OF PARAMETERS

We show C-NODE's parameter efficiency over NODE with an ablation study on the image classification task on the CIFAR-10 dataset. Specifically, under a similar training setup, we experiment with C-NODE with 95071, 55855, and 17379 parameters and experiment with NODE with 96044, 56828, and 17444 parameters. As shown in Figure 15, although C-NODE has more variance in its performance, it outperforms NODE along the whole training process in all three cases.

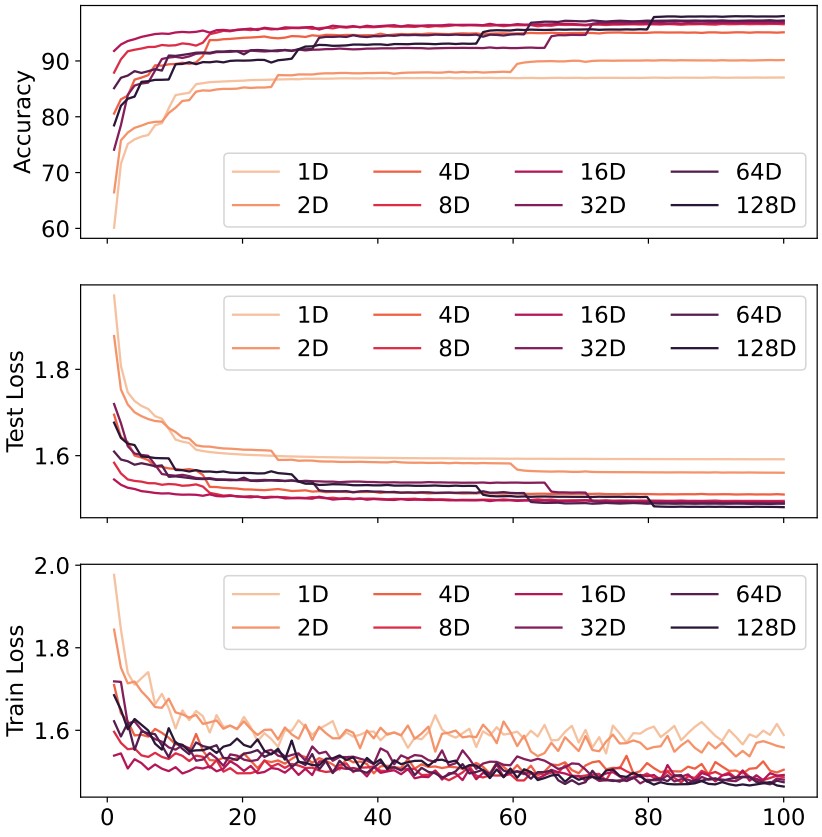

Figure 12: The training process averaged over 4 runs of C-NODE with 1, 2, 4, 8, 16, 32, 64, 128, 256, 512, and 1024 dimensions on the MNIST dataset. The first row is the accuracy of prediction, the second row is the testing error, and the third row is the training error.

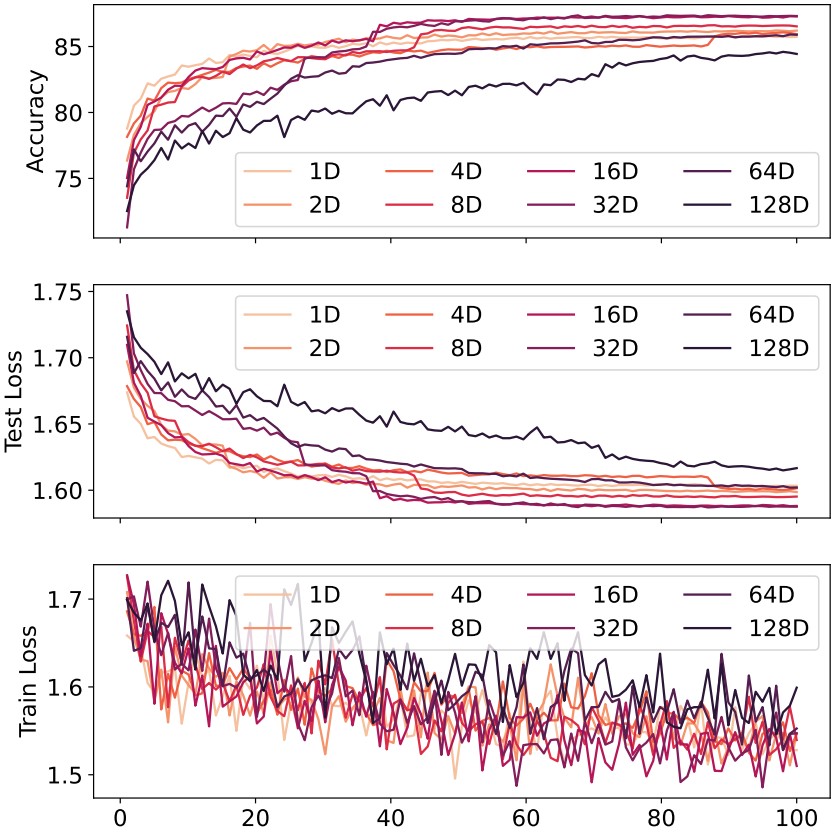

Figure 13: The training process averaged over 4 runs of C-NODE with 1, 2, 4, 8, 16, 32, 64, and 128 dimensions on the SVHN dataset. The first row is the accuracy of prediction, the second row is the testing error, and the third row is the training error.

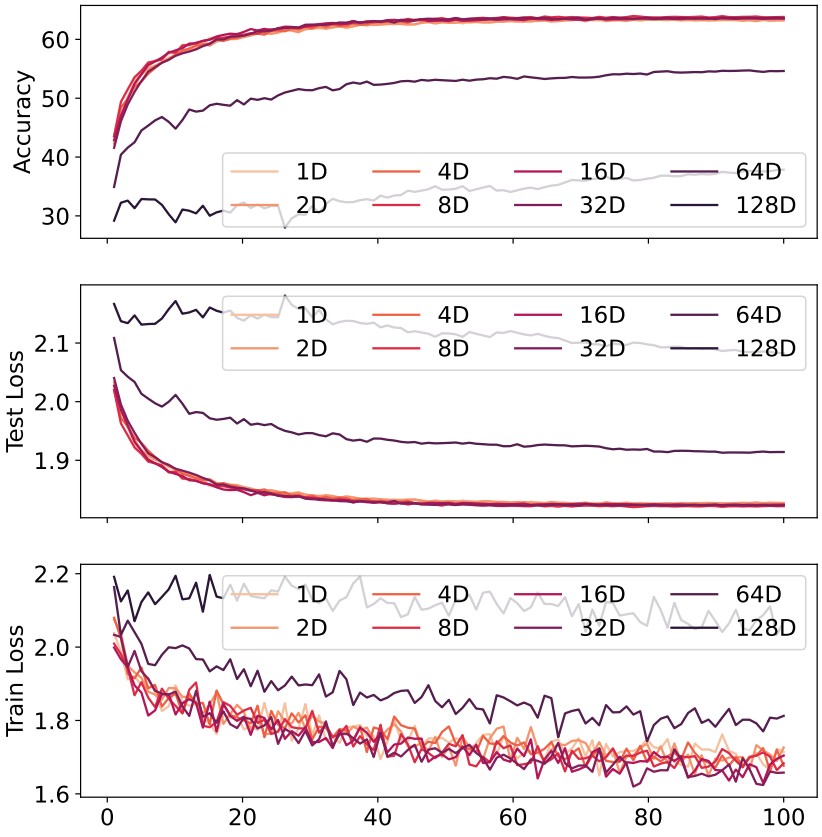

Figure 14: The training process averaged over 4 runs of C-NODE with 1, 2, 4, 8, 16, 32, 64, and 128 dimensions on the CIFAR-10 dataset. The first row is the accuracy of prediction, the second row is the testing error, and the third row is the training error.

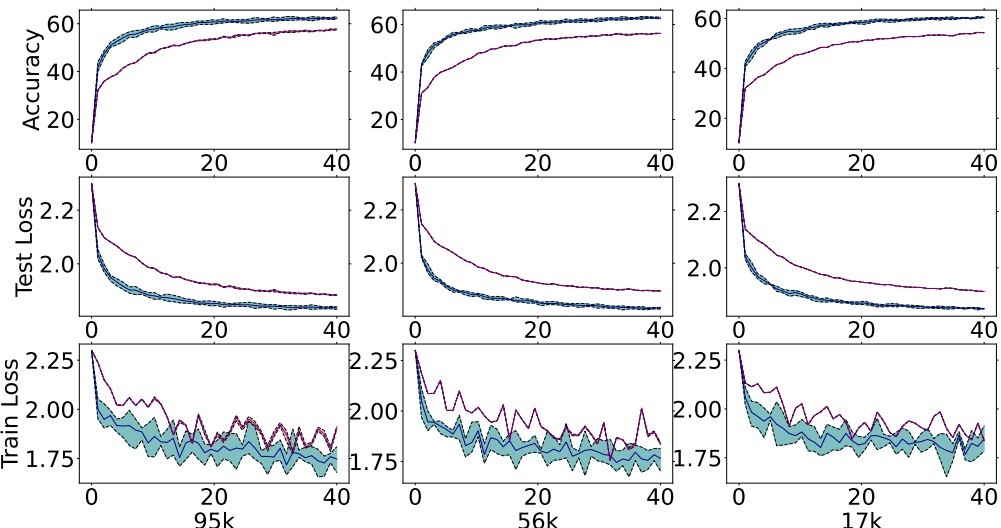

Figure 15: The training process averaged over four runs of C-NODE with 95071, 55855, and 17379 parameters on the CIFAR-10 dataset, and NODE with 96044, 55855, and 17379 parameters. The first row is the prediction accuracy, the second row is the testing error, and the third row is the training error. Blue lines are the results for C-NODE, and red lines are the results for NODE.

---

**Algorithm 2** C-NODE algorithm using the forward Euler method

---

**for** each input data $\mathbf{z}_j$ **do**
    extract image feature $\mathbf{u}(s = 0) = \mathbf{g}(\mathbf{z}_j; \Theta_1)$ with a feature extractor neural network.
    **procedure** Integration along $s = 0 \rightarrow 1$
    **for** each time step $s_m$ **do**
        calculate $\frac{d\mathbf{x}}{ds}(\mathbf{x}, \mathbf{u}; \mathbf{g}(\mathbf{z}_j; \Theta_1); \Theta_2)$ and $\mathbf{J_x u}(\mathbf{x}, \mathbf{u}; \Theta_2)$.
        calculate $\frac{d\mathbf{u}}{ds} = \mathbf{J_x u} \frac{d\mathbf{x}}{ds}$.
        calculate $\mathbf{u}(s_{m+1}) = \mathbf{u}(s_m) + \frac{d\mathbf{u}}{ds}(s_{m+1} - s_m)$.
        calculate $\mathbf{x}(s_{m+1}) = \mathbf{x}(s_m) + \frac{d\mathbf{x}}{ds}(s_{m+1} - s_m)$
    **end for**
    **end procedure**
    classify $\mathbf{u}(s = 1)$ with neural network $\Phi(\mathbf{u}(\mathbf{x}(s = 1)), \Theta_3)$.
**end for**

---

# F ALGORITHM FOR IMAGE CLASSIFICAITON WITH C-NODE

We provide algorithm for training C-NODE for the purpose of image classification.

# G ALGORITHM FOR CONTINUOUS NORMALIZING FLOWS DEFINED WITH C-NODE

We additionally provide algorithms for training and sampling CNFs defined with C-NODEs.

---

**Algorithm 3** Algorithm for training CNFs defined with C-NODE

---

given probability density function of $p(\mathbf{u}(s = 0)) = p_0(\cdot)$
**for** each input data $\mathbf{z}_j$ **do**
    Given $\begin{bmatrix} \mathbf{u}(1) \\ \log p(\mathbf{z}_j) - \log p(\mathbf{u}(1)) \end{bmatrix} = \begin{bmatrix} \mathbf{z}_j \\ 0 \end{bmatrix}$

    **procedure** Integrate from $1 \rightarrow 0$ to get $\begin{bmatrix} \mathbf{u}(0) \\ \log p(\mathbf{z}_j) - \log p(\mathbf{u}(0)) \end{bmatrix}$

    **for** each time step $s_m$ **do**
        calculate $\frac{d\mathbf{x}}{ds}(\mathbf{x}, \mathbf{u}; \mathbf{g}(\mathbf{z}_j; \Theta_1); \Theta_2)$ and $\mathbf{J_x u}(\mathbf{x}, \mathbf{u}; \Theta_2)$.
        calculate $\frac{d\mathbf{u}}{ds} = \mathbf{J_x u} \frac{d\mathbf{x}}{ds}$.
        calculate $-\mathrm{tr}(\frac{\partial}{\partial \mathbf{u}} \mathbf{J_x u} \frac{d\mathbf{x}}{ds})$ with Hutchinson trace estimator (Grathwohl et al., 2019).
        calculate $\begin{bmatrix} \mathbf{u}(s_{m+1}) \\ \log p(\mathbf{u}(s_{m+1})) \end{bmatrix} = \begin{bmatrix} \mathbf{u}(s_m) \\ \log p(\mathbf{u}(s_m)) \end{bmatrix} + \begin{bmatrix} \frac{d\mathbf{u}}{dt} \\ \frac{\partial \log p(\mathbf{u}(s))}{\partial s} \end{bmatrix} (s_{m+1} - s_m)$.
    **end for**
    evaluate $p_0(\mathbf{u}(0))$
    calculate $\log p(\mathbf{z}_j) = (\log p(\mathbf{z}_j) - \log p(\mathbf{u}(0))) + \log p_0(\mathbf{u}(0))$
    optimize $\log p(\mathbf{z}_j)$ with an optimization algorithm (stochastic gradient descent etc.)
**end for**

---

---

**Algorithm 4** Algorithm for sampling CNFs defined with C-NODE

---

**procedure** sample $\mathbf{u}(s = 0)$ from base distribution $p_0(\cdot)$
**procedure** Integrate from $0 \rightarrow 1$ to get $\mathbf{u}(s = 1)$
**for** each time step $s_m$ **do**
    calculate $\frac{d\mathbf{x}}{ds}(\mathbf{x}, \mathbf{u}; \mathbf{g}(\mathbf{z}_j; \Theta_1); \Theta_2)$ and $\mathbf{J_x}\mathbf{u}(\mathbf{x}, \mathbf{u}; \Theta_2)$.
    calculate $\frac{d\mathbf{u}}{ds} = \mathbf{J_x}\mathbf{u}\frac{d\mathbf{x}}{ds}$.
    calculate $\mathbf{u}(s_{m+1}) = \mathbf{u}(s_m) + \frac{d\mathbf{u}}{ds}(s_{m+1} - s_m)$.
**end for**
**end procedure**
$\mathbf{u}(s = 1)$ is our sample from the CNF

---

