# OpenReview forum: "Characteristic Neural Ordinary Differential Equation"
_ICLR.cc/2023/Conference — ICLR 2023 poster_

### Official Review · Reviewer_eEjp · 2022-10-23

**Confidence:** 4
**Correctness:** 3
**Technical Novelty And Significance:** 3
**Empirical Novelty And Significance:** 3
**Recommendation:** 6

**Clarity, Quality, Novelty And Reproducibility:**

### Clarity

The paper is very clear. I believe some improvements can be made in the intuition why the space of dynamics is richer in C-NODE (see my comment above about the interesecting trajectories). Clarifications regarding the inconsistency between invertibility and interesecting trajectories is also needed.

### Quality

The work is qualitative, and the experimental section is broad and convicing.

### Novelty

The novelty comes from the method of characteristics. I believe this would deserve some extra experiments to showcase the value of this theoretical framework (see my comment above).

### Reproducibility

The derivations of the model are clear and should allow for reproducing the experiments.

**Details Of Ethics Concerns:**

I have no ethical concerns.

**Strength And Weaknesses:**

## Strengths

- The method is nicely presented and the overall paper is clear. The method is also well motivated, using the method of characteristics for solving PDEs.

- The experimental section is solid, with various setttings (spanning from classification, to time series modeling and normalizing flows). The results indicate improved performance of C-NODE.

## Weaknesses


From the paper, I read two motivations : the ability to keep the adjoint method for training and the ability to model more complex dynamics than NODE (in particular intersecting trajectories). Yet, I think these are currently under-developped in the paper.

*About the adjoint*

- For the adjoint method, from experience, the improved memory complexity comes at the cost of significantly higher computational time, making it therefore unpractical. Therefore, this motivation is quite weak in my opinion. What is more, the cases used in the experimients don't seem to be requiring the adjoint. method for training either. I would therefore recommend to make this adjoint argument more compelling by showing a case study where the adjoint is indeed necessary.

*About the ability to model intersecting trajectories*

- The ability to model intersecting trajectories seem to arise from the fact that you include the first time point $g(\mathbf{z})$ in the input of the dynamical system function. In my sense, that alone suffices to allow for intersecting trajectories. If you agree, this comes, in my sense, with multiple implications. First, that statement should be made more clear in the text so the reader can better see where this ability comes from. Second, this would represent the most interesting baseline to compare against. Namely, you take a neural ODE and you feed the first $\mathbf{z}$ as input, similiarly as you do for the dynamics of the characteristics. This would be, in my opinion, a more compelling argument for using the method of characteristics framework.
- What is more, the non-intersecting property clashes with the invertibility statement of section 4.2. Indeed, I do not see how intersecting can be invertible as they won't be injective. I think this would deserve some clarification. I believe this confusion is due to what you consider is the input (if you condition on $z$, it is invertible$, but not otherwise).



**Summary Of The Paper:**

This paper proposes to model the evolution of latent variables in continuous time models as the solution of first-order partial differential equations. The authors builds upon the method of characteristics for solving PDEs and show that their approach provides both theoretical and practical advantages. In particular, they show that their method can model interesecting trajectories and can still benefit from the adjoint method for backpropagation. Experimentally, the method outperforms classical NODE and ANODE in a variety of tasks (classificiation, time series classification as well as NF).

**Summary Of The Review:**

This work proposes and interesting idea from PDE solving to improve NODE architecture. The experiments are broad in scope. I still have some concern regarding a mismatch between the experiments and the motivations of the paper : regarding the adjoint and the usage of the first input in the dynamics. Some work on the motivation and comparison with more relevant baseline (the one I suggested above) would be appreciated.

---

> ### Author Response · Authors · 2022-11-14
> **Comment Reply**
>
> We thank the reviewer for the insightful comments and particularly relevant suggestions for improving the paper.
> We address individual concerns below.
>
> - [Question 1: Adjoint motivation]
>
> The reviewer brings up an interesting point.
> Indeed, the adjoint method results in an increased computation time, and in many cases directly backpropagating through the solver is sufficient.
> However, for certain tasks, using the adjoint method is necessary to compute the integrals due to the numerical errors and memory costs incurred from using standard backpropagation through the integrator.
> In particular, in our experiments on continuous normalizing flows, the numerical error from a standard Euler solver often results in an undefined log-likelihood.
> Using an adaptive solver, while more costly, removes this error.
> We have modified the manuscript to bring this point to light.
>
> - [Question 2: Intersecting trajectories]
>
> We apologize for the confusion regarding invertibility and intersecting dynamics.
> For each individual characteristic, the characteristic provides an invertible map over the domain.
> When grouping many of these characteristics (as the reviewer mentions, with a different initial condition) we can obtain intersecting dynamics.
> The continuous normalizing flows case applies the conditional characteristic to the initial distribution to form the target density, leading to the invertibility along that characteristic.
> We have modified the manuscript to include these points as a clarification.
>
> - [Question 3: Baseline of a conditional NODE]
>
> We thank the reviewer for bringing up this interesting idea for experimentation.
> We follow the reviewer's suggestion and compute the accuracy of the classification tasks compared to a conditional NODE and the C-NODE formulation.
> Notably, the NODE formulation could be seen as solving a series of ODEs conditioned on the data whereas C-NODE conditions the characteristics on data over an arbitrary dimensional space.
> We added Figure 11 in Appendix C to illustrate the differences.
> C-NODE outperforms the conditional NODE in all datasets when considering the normal backpropagation through Euler, likely due to the influence of the learned characteristics providing more amenable dynamics.
> C-NODE also outperforms conditional NODE in both SVHN and CIFAR-10 when using the adjoint training, but both perform similarly in MNIST.
> We believe this further validates the importance of using the characteristics, and we thank the reviewer again for suggesting this experiment.

---

> > ### Comment · Reviewer_eEjp · 2022-12-06
> > **Thank you for your responses**
> >
> > Thank you for your responses.
> >
> > Regarding the adjoint, should I understand that in practice, using an adaptive solver is enough and that the adjoint is not necessary ?
> >
> > Regarding the clarification for the ability to model intersecting trajectories. I can't find updates about this in the new version of the paper. Based on your answer, it seems correct to say that the intersection properties comes from the fact that you feed the first time point $g(z)$ in the dynamics of the system. If correct, this is not clear from the current version of the paper. I believe it is worthwile conveying more evidently the inner workings of the method.
> >
> > Regarding the comparison with baseline. Thank you for running this experiment. A table with quantitative results would have been appreciated as the figures are not very readable for comparison.

---

> > > ### Author Response · Authors · 2022-12-07
> > > **Thank you for the suggestions**
> > >
> > > Thank you for the questions.
> > >
> > > •  [Question 1: Usage of adaptive solver]
> > >
> > > The reviewer is correct that using an adaptive solver would work in practice when the data is of low dimensions or when the number of function evaluations remains small. However, when handling high-dimensional data (e.g. images), a regular adaptive solver might take too much memory. Adaptive solvers take memory on the order of $\mathcal{O}(\tilde{L})$, where $\tilde{L}$ is the number of function evaluations, whereas the adjoint method takes memory on the order $\mathcal{O}(1)$, and is independent to the number of steps taken by adaptive solvers.
> > >
> > > •  [Question 2: More explanations on conditioning on initial condition]
> > >
> > > The reviewer is correct that the ability to model intersecting trajectories arises by imposing a dependence on the characteristics of the data point.
> > > We mentioned this in the "conditioning on data" paragraph.
> > > However, we will add the following modifications to make this property more apparent.
> > >
> > > In the "conditioning on data" paragraph, we will note that:
> > > This property becomes helpful in Proposition 4.1 in proving that C-NODE can represent intersecting dynamics.
> > >
> > > We also will edit the paragraph on intersection trajectories in section 4 to include:
> > >
> > > As mentioned in Dupont et al. (2019), one limitation of NODE is that the mappings cannot represent intersecting dynamics.
> > > We prove by construction that the C-NODEs can
> > > represent some dynamical systems with intersecting trajectories.
> > > The key ingredient in having C-NODE represent intersecting trajectories is to modify the initial condition of the characteristics in an appropriate manner.
> > > Modifying the initial condition then gives additional flexibility in the modeling framework.
> > > We formalize this in the following proposition:
> > >
> > > Finally, in Appendix C, "Including initial state in NODE's input", we will add the following:
> > >
> > > By conditioning the characteristic curves on initial values, the method can model dynamical systems with
> > > intersecting trajectories. C-NODE and conditioned NODE are two different methods of including the initial
> > > values.
> > >
> > > •  [Question 3: Table with quantitative results]
> > >
> > > Thanks for the suggestion. We initially included a table on the parameter counts for different models in the conditional-NODE experiment (Table 20), and we will replace Table 20 with the following table:
> > > | Dataset  | Method | Accuracy (Euler) $\uparrow$    | Accuracy (Adaptive) $\uparrow$ | Param. [K] $\downarrow$ |
> > > |----------|--------|---------------------|---------------------|------------|
> > > | SVHN     | NODE   | $82.42\pm 0.043\%$  | $64.510\pm 1.637\%$ | 115.444    |
> > > |          | COND   | $83.30\pm0.131\%$   | $80.39\pm1.088\%$   | 116.399    |
> > > |          | C-NODE | $86.28\pm0.086\%$   | $82.35\pm1.458\%$   | 113.851    |
> > > | CIFAR-10 | NODE   | $59.492\pm 0.366\%$ | $52.13\pm1.464\%$   | 115.444    |
> > > |          | COND   | $56.77\pm0.137\%$   | $57.71\pm0.226\%$   | 118.447    |
> > > |          | C-NODE | $61.97\pm0.139\%$   | $63.57\pm0.421\%$   | 113.851    |
> > > | MNIST    | NODE   | $97.09\pm0.024\%$   | $96.79\pm0.374\%$   | 85.468     |
> > > |          | COND   | $93.38\pm1.798\%$   | $98.05\pm0.112\%$   | 87.287     |
> > > |          | C-NODE | $96.93\pm0.103\%$   | $97.26\pm0.276\%$   | 83.041     |

---

### Official Review · Reviewer_TVDi · 2022-10-23

**Confidence:** 2
**Correctness:** 3
**Technical Novelty And Significance:** 3
**Empirical Novelty And Significance:** 2
**Recommendation:** 5

**Clarity, Quality, Novelty And Reproducibility:**

**Clarity:**
The description of the method could be improved overall. To me – being not an expert on the method of characteristics – it is not entirely clear how the proposed method works. In particular, it is not clear how the coupled ODE (4)&(5) is solved in praxis: I assume that in general one will solve iteratively for the solution in both $x$ and $\symbf u$ simultaneously, however, in Algorithm 1, not update of $x$ appears. Further, the domain of $u$ is not entirely clear: it is introduced as a function on $\mathbb R^k$, I assume it should be a function of $\mathbb R^k\times\mathbb R$ and later it (or rather its Jacobian) has arguments $(x, \symbf u)$. Similar inaccuracies are common in the manuscript, which makes the definition of the model hard to follow. Overall, I believe the manuscript requires some polishing in order to improve readability.

**Quality:**
The paper provides a nice extension of NODEs and I appreciate the inspiration from a method from PDE theory. The experiments -- apart from Subsection 5.4 -- are well designed and compare the method in a variety of settings to various baselines.


**Novelty:**
The idea is intriguing and appears novel to me. That being said, I am not an expert on NODEs.

**Reproducability:**
The reproducability could be improved. For example, for the classification problem, neither the loss nor the architecture nor the hyperparameters are reported. Further, the code is not available.


**Strength And Weaknesses:**

**Strengths:**
* The proposed model is based on an important theoretical tool in the theory of PDEs.
* The expressivity of the proposed model is studied theoretically
* The experiments showing performance gains of the model cover a variety of different problems and include comparisons against a variety of different methods.

**Weaknesses:**

* On the side of the experiments, I believe the computation time should be added for a reference, since the number of parameters does not completely describe the complexity of training. Further, in the limitations it is stated that the proposed model exhibits instabilities during training. A discussion regarding this should already present in the experiments section.

* The experiment on the representational power when it comes to approximating solutions of PDEs seems not well desgined. As far as I see it, C-NODEs are shown to exhibit lower generalization error compared to NODEs when used to regress the solution of a hyperbolic PDE. This however does not imply that C-NODEs are more effective in approximating the regressed function.

* One weakness of the paper is the motivation to use a model inspired by hyperbolic PDEs. The main argument given is its ability to „transport data into different regions in state space“, which is also highlighted by the theoretical expressivity results. Where I agree that this is indeed a favorable property, this is already the case for simple augmented NODEs. To me it is not clear whether the improvements in the experiments are actually due to an underlying hyperbolic structure in the data of the experiments. That being said, I think this is fine, since the manuscript only speculates about this and does not offer this as an explanation.


**Summary Of The Paper:**

The manuscript introduces a neural ODE (NODE) type model inspired by the method of characteristics from PDE theory, which is therefore referred to as characteristic NODE or C-NODE. Compared to vanilla NODEs the resulting parametric dynamical system can exhibit intersecting dynamics, which can be desirable for certain applications.
This result is strengthened by showing that the proposed model is more expressive than vanilla ODEs in the sense that it can compute arbitrary homeomorphisms as input-output maps.
Further, a formula for the evoluation of the log density under C-NODEs is presented, which makes the model applicable to density estimation problems.
The proposed model is tested on image classification, time series prediction, density estimation as well as approximation of the solution of a quasilinear hyperbolig PDE.


**Summary Of The Review:**

The manuscript proposes an interesting new NODE type model, which is inspired by the method of characteristics for hyperbolic PDEs.
Overall, I believe that the idea is intriguing and promising. In its current form, I am mostly concerned about the clarity of the paper (the introduction of the model is sloppy or at least not very clear) as well as the reproducability. I believe that both of these points should be addressed prior to a publication of the manuscript.

---

> ### Author Response · Authors · 2022-11-14
> **Comment Reply (Part 1)**
>
> We thank the reviewer for the review and comments. We address individual concerns below.
>
> - [Question 1: Adding computation time]
>
> We agree that computational times are helpful, but unfortunately we did not record them when we initially ran the experiments.
> Due to the limited time available in the response period, it is difficult to retrain all the models by the deadline.
> For this response period, we include plots comparing the integration time of NODE and C-NODE in the updated manuscript in Figure 9 of the appendix to demonstrate this on the PDE solving example.
> The plots in Figure 9 suggest that C-NODE and NODE have approximately the same integration time in the adjoint method and slightly larger computation time using the forward Euler method.
> We note that this may be improved with some optimization of the computations in C-NODE (e.g. of the matrix multiplication), but considering this is beyond the scope of the manuscript.
> We also note that we have included the number of function evaluations in the original manuscript, and these should roughly correspond to the integration time of the different methods.
> The tables suggest C-NODE tends to have fewer function evaluations than the relevant baselines.
> We will nevertheless include additional computation times in the final revision given sufficient time.
>
> - [Question 2: Instabilities during training]
>
> We agree with the reviewer that the instability should be further investigated.
> We provide additional explanation on instability and how it manifests in the different NODE methods and note that it is a fairly minor occurrence.
> In our experiments, we found that training becomes unstable when the number of function evaluations exceeds 1000, which is how we are defining instability.
> This mostly occurred when training on the SVHN dataset, and we modified the experimental section and discuss this in greater detail.
>
> To empirically investigate this further, we provide an additional experiment to compare the instability by counting the number of times each method exceeds 1000 function evaluations within the training period.
> When training NODE, C-NODE, and ANODE for image classification on the SVHN dataset for forty instances, NODE appeared unstable six times, C-NODE was unstable three times, and ANODE was never unstable.
> However, when considering the average number of function evaluations in Table 1, C-NODE tends to reduce the average number of function evaluations, suggesting that C-NODE has a stabilizing effect during optimization.
> Additionally, when training the ANODE+C-NODE experiments, the training procedure was never unstable.
> We have included the following paragraph in the experiments section:
> "While the average number of function evaluations tends to be lower for C-NODE, we additionally note that, compared to ANODE, training C-NODE with the adjoint method can sometimes have decreased stability. We define instability as having an NFE $>1000$. To get a rough idea of the differences in stability, when training NODE, C-NODE, and ANODE for image classification on the SVHN dataset for forty instances, NODE appeared unstable six times, C-NODE was unstable three times, and ANODE was never unstable. We note that this was only apparent in the SVHN experiment and when considering C-NODE by itself; the average NFE decreases when adding C-NODE to A-NODE and it was never experienced in the ANODE+C-NODE experiments."

---

> ### Author Response · Authors · 2022-11-14
> **Comment Reply (Part 2)**
>
> - [Question 3: Improper evaluation of PDE experiments]
>
> We note that computing the MSE on held-out data is the general method of validating different numerical solvers, see for example:
>
> [1] Raissi, M., Perdikaris, P., \& Karniadakis, G. E. (2019). Physics-informed neural networks: A deep learning framework for solving forward and inverse problems involving nonlinear partial differential equations. Journal of Computational physics, 378, 686-707.
>
> [2] Yu, B. (2018). The deep Ritz method: a deep learning-based numerical algorithm for solving variational problems. Communications in Mathematics and Statistics, 6(1), 1-12.
>
> [3] Sirignano, J., \& Spiliopoulos, K. (2018). DGM: A deep learning algorithm for solving partial differential equations. Journal of computational physics, 375, 1339-1364.
>
> [4] Han, J., Jentzen, A., \& E, W. (2018). Solving high-dimensional partial differential equations using deep learning. Proceedings of the National Academy of Sciences, 115(34), 8505-8510.
>
> We followed a similar procedure in our validation, since it also applies to our experiments.
> However, we would be happy to consider an alternative metric that the reviewer sees fit for this experiment.
>
> - [Question 4: Unclear motivation of the model]
>
> The reviewer is correct that the transport interpretation is based on the theoretical properties of hyperbolic PDEs.
> All experimental procedures control for the number of parameters to minimize the influence of factors unrelated to the main contributions of the proposed method.
> Additionally, we combined the C-NODE method with existing architectures to understand the influence of this particular component on the performance.
> In that sense, the experiments are designed to explicitly isolate the effect of the C-NODE architecture as best as possible.
>
> - [Question 5: Improved description of the method]
>
> We thank the reviewer for pointing this out, and we have modified the manuscript to correct the inaccurate notations.
>
> - [Question 6: Reproducibility]
>
> We would like to note that the code has been available in the supplementary material since the initial submission.
> We additionally included specific information on the experiments in the revised appendix, with emphasis on the points that the reviewer requested.

---

### Official Review · Reviewer_74Jk · 2022-10-25

**Confidence:** 4
**Correctness:** 4
**Technical Novelty And Significance:** 4
**Empirical Novelty And Significance:** 3
**Recommendation:** 6

**Clarity, Quality, Novelty And Reproducibility:**

Quality:
The paper is of high quality, with novel idea and solid theory.

Clarity:
The paper is well presented and clear.

Originality:
The paper is novel and pretty original

**Strength And Weaknesses:**

Strength:
1. The idea to extend Neural ODE to PDE via Characteristic equations is novel. It great expands the application of Neural ODEs.
2. The theoretical derivation is solid. (Though I only loosely checked the proofs the theories, they look pretty convincing to me).
3. The paper is well-presented and easy to follow, especially in the method section, the authors started from the general background of characteristic equations for PDE, then extend them to the Neural ODE case.

Weakness:
1. Could the authors expand a bit more on the difference between augmented NODE and C-NODE? The formulation of C-NODE essentially models the original NODE in a higher dimension, which looks close to augmented NODE. So I wonder what is the key difference.
2. Experiments
    Despite the nice theory and presentation, the paper lacks comparison with some stronger baselines. For example, [1,2,3] achieved slightly better results on CNF on Cifar10, though they are focused on numerical solvers rather than PDE, it's worth mentioning in the paper.

PS: I doubt if the slightly worse result of C-NODE is because the model is smaller than in the literature. I would be happy to increase score if the author show better results with a model of comparable size.

[1] Zhuang, Juntang, et al. "Adaptive checkpoint adjoint method for gradient estimation in neural ode." International Conference on Machine Learning. PMLR, 2020.
[2] Matsubara, Takashi, Yuto Miyatake, and Takaharu Yaguchi. "Symplectic adjoint method for exact gradient of neural ode with minimal memory." Advances in Neural Information Processing Systems 34 (2021): 20772-20784.
[3] Zhuang, Juntang, et al. "MALI: A memory efficient and reverse accurate integrator for Neural ODEs." arXiv preprint arXiv:2102.04668 (2021).

**Summary Of The Paper:**

The authors proposed Characteristic Neural ODE (C-NODE), which extends the idea of Neural ODE to combine with PDEs, and enables the model to learn a family of models (rather than a single Neural ODE model).

Furthermore, the authors demonstrated that C-NODE has a nice property, that it can model intersection trajectories (similar to augmented Neural ODE), in contrast Neural ODE losing such capability. The authors also derived the formulation for continuous normalizing flow with C-NODE (in the PDE settings).

Finally, the authors validated the performance of C-NODE with experiments.

**Summary Of The Review:**

The paper is novel, well presented and solid in theory. Though a little bit weak in experiment, but it could come from that the model size is small. Overall is a very good paper.

---

> ### Author Response · Authors · 2022-11-14
> **Comment Reply**
>
> We thank the reviewer for the review and the helpful comments regarding the manuscript. We address individual concerns below.
>
> - [Question 1: Discussion on the differences between augmented NODE and C-NODE]
>
> Augmented NODE considers a higher dimensional latent space whereas C-NODE retains the same dimensional latent space but considers the transport over a possibly higher dimensional space (the space of the characteristics). This makes C-NODE amenable for tasks such as continuous normalizing flows, which would not be naturally defined in the case of an augmented dimension.
> Additionally, C-NODE conditions these characteristics on the data, which has the effect of solving according to a different characteristic, motivating the interpretation that the union of the solutions over different characteristics solves the PDE.
>
> - [Question 2: Comparing with stronger baselines]
>
> We thank the reviewer for pointing these baselines out to us.
> We have revised the manuscript and included a discussion on them in the revision in the related work section.
> From our understanding, the papers suggest that these gains are largely from the numerical integrators that are proposed in each paper rather than the properties of the architecture.
> We speculate that adding these integrators to C-NODE could improve the performance as well, but we did not have time to complete this experiment.
> We did attempt an experiment on MNIST with a larger C-NODE neural network (with 850k parameters) based on the codebase of [1].
> From our experiment, the average Bits/Dim is 0.824, with a standard error of 0.0232. We used the dopri5 solver in the Torchdiffeq package, with both the relative tolerance and the absolute tolerance being 1e-5. This is comparable with the Bits/Dim of 0.87 previously achieved in [1], where they used a model with 800k parameters.
>
> [1]Juntang Zhuang, Nicha C Dvornek, sekhar tatikonda, and James s Duncan. {MALI}: A memory
> efficient and reverse accurate integrator for neural {ode}s. In International Conference on Learning
> Representations, 2021.

---

### Official Review · Reviewer_212J · 2022-10-30

**Confidence:** 3
**Correctness:** 4
**Technical Novelty And Significance:** 4
**Empirical Novelty And Significance:** 3
**Recommendation:** 8

**Clarity, Quality, Novelty And Reproducibility:**

This work introduces a novel method for PDE approximation. The presented ideas and theory are sound and interesting. Overall, the paper is of high quality, though clarity can be slightly improved by streamlining the presentation and organization of the paper.

**Strength And Weaknesses:**

This paper introduces a novel and relevant extension for the existing Neural ODE framework, and the presented experimental results are significant. The proposed framework allows to model a broad class of interesting PDEs, i.e., hyperbolic PDEs. I don’t feel that it is a restrictive limitation of the framework that it can handle only hyperbolic PDEs, since this class is of general interest in many problems.

While the key innovation of this paper is a new deep learning method for PDE approximation, my concern is that this framework could just be viewed as a version of augmented NODEs. However, the experiments seem to suggest that the proposed method gives good results. To me, the strength here is in the interpretation, but I'm guessing from the PDE experiment that you cannot get the NNs to represent what their analytical solution counterparts should be. Still, I think that this paper is a good step in the right direction.  Here are things that I would like to see addressed, and that I feel will help to improve the paper:


* The presentation of the mathematical exposition can be streamlined in several places to make the paper better readable. I feel that sometimes some unnecessary terms/notations are introduced without additional explanation. For instance, the middle term in the first equation of the related work is more confusing than helpful. I don’t get why it is important to explicitly state that the right-hand side is a function of (u,t). If this notation is important to understand later concepts, then it should be mentioned / discussed with a sentence within the text. I would try to keep things simple so that the paper is more accessible to a broader audience.


* The related work is discussing that a residual unit can be modeled as an ODE in the limit. While in theory that is fine, this statement is problematic in practice. Typically, you can’t choose $\Delta t$ arbitrarily small, if you want to train a model in finite time. Hence, while there is some algebraic resemblance, the connection between resnets and the forward Euler discretization of an ODE is problematic. I would suggest citing works that point out some of these issues: [1,2].


* Since the presented approach is a method for PDE approximation, I would suggest devoting more room for experimental results in the main body of the manuscript to illustrate the framework, i.e., move some of the experiments from the appendix to the main text and expand the discussion. Additional experiments for PDEs in the appendix would be welcome. I am saying this, because I don’t feel that the proposed method is of interest for image classification. Also, the image classification experiments ignore mention of state-of-the-art Neural ODE frameworks [3,4]. For instance [4] achieves about 90% test accuracy on CIFAR-10 with a similar parameter count, and 99.6% test accuracy on MNIST with only 18k parameters. These works should at least be cited in the related works.


* The quality of the plots can be improved. Instead of having solid confidence bands, I suggest using transparent confidence bands so that it can be seen how large the overlap is.


* Lastly, I would add a short discussion about the differences / advantages of the proposed approach compared to physics-informed neural networks. I think that the proposed approach is more general for modeling PDEs. So, it would be nice to point this out.



[1] Ott, Katharina, et al. "Resnet after all: Neural odes and their numerical solution." International Conference on Learning Representations. 2020.

[2] Krishnapriyan, Aditi S., et al. "Learning continuous models for continuous physics." arXiv preprint arXiv:2202.08494 (2022).

[3] Zhang, Tianjun, et al. "ANODEV2: A coupled neural ODE framework." Advances in Neural Information Processing Systems 32 (2019).

[4] Queiruga, Alejandro, et al. "Stateful ODE-Nets using basis function expansions." Advances in Neural Information Processing Systems 34 (2021): 21770-21781.

**Summary Of The Paper:**

Partial differential equations (PDEs) are ubiquitous in science and engineering, and modern deep learning frameworks for modeling PDEs are of interest for scientific machine learning. This paper leverages the method of characteristics to extend the framework of Neural Ordinary Differential Equations (Neural ODEs) to modeling (hyperbolic) PDEs. The method of characteristics allows to transform hyperbolic PDEs into ODEs. Then, tools for learning Neural ODEs can be used to model the transformed PDE. Several different problems are used to demonstrate the performance of the proposed Characteristic Neural ODE framework.

**Summary Of The Review:**

In summary, the presented ideas are solid, and I feel that the innovations are novel and significant. The proposed framework connects Neural ODEs to an interesting class of PDEs that are relevant in many real-world applications. Hence, I have no doubt that this work has the potential to inspire future research in this area. Overall, this paper is of comparable quality compared to several related works that have been recently been accepted at ICLR and other ML conference.  Given that the reviewers address my comments, I recommend to accept this paper.

---

> ### Author Response · Authors · 2022-11-14
> **Comment Reply**
>
> We thank the reviewer for the constructive comments and the time for reviewing the manuscript.
> We address individual comments below.
>
> - [Question 1, Improved presentation of mathematical notation]
>
> Following the reviewer's suggestion, we have improved the mathematical notation in a few places within the manuscript.
> We additionally streamlined the notation of the characteristic to write it as
> $$
>     u(t_1) = u(t_0) + \int_{t_0}^{t_1} \frac{d u(s)}{ds} \mathrm{d} t = u(t_0)+ \int_{t_0}^{t_1} f(u(s), s, \theta) \mathrm{d} s
> $$
>
> - [Question 2, Relationship between ResNets and Neural ODEs]
>
> We agree with the reviewer, the interpretation of neural ODEs as a continuous ResNet is only an inspiration, and various works suggest that the relationship is not empirically supported.
> We included the following description in the related work:
>
> "The interpretation of NODE as a continuous form of ResNet is also problematic, owing to the fact that the empirical behavior of the ResNet does not match the theoretical properties as studied in Krishnapryan 2022, Ott 2021."
>
>
> - [Question 3, Include additional PDE experiments]
>
> We agree with the reviewer that these are some of the more interesting experiments.
> To strengthen the analysis on learning PDEs, we added another experiment on a 100 dimensional transport equation which is available in Appendix A.4.2.
>
> - [Question 4, Additional references on NODE variants]
>
> Thank you for pointing these papers out to us.
> We included a discussion on these methods in the related work section.
> Specifically, we include the following:
>
>
> "Additional interpretations of the process represented by ODE have been considered.
> In Zhang 2019, the authors considered an augmentation where the augmented state corresponds to the parameters of the network governing the latent state, rather than the usual augmentation directly on the latent state.
> In another vein, Queiruga 2021 describes the latent evolution through a series of basis functions thereby allowing important concepts such as BatchNorm to be effectively translated in the continuous setting, achieving state of the art performance on a variety of image classification tasks. "
>
> - [Question 5, Plot quality]
>
> We thank the reviewer for the suggestion, and we have updated the plots to include transparent confidence bands.
>
> - [Question 6, Comparison to Physics Informed Neural Networks]
>
> The reviewer brings up a good point, and we have included a note in the revision in Appendix A.4.2.
> There are several differences between the proposed CNODE and the PINNs framework.
> Mainly, we outline that PINNs use a regularization-based approach that requires the exact form of the PDE but can be applied to a very general range of PDEs.
> However, PINNs do not typically scale well in higher dimensions due to the difficulties in computing some derivatives in higher dimensional spaces.
> On the other hand, the C-NODE approach works only for the class of hyperbolic PDEs, but does not require knowledge of the exact form of the PDE and scales to high-dimensional equations.

---

> > ### Comment · Reviewer_212J · 2022-12-06
> > **Thanks**
> >
> > I would like to thank the authors for the detailed response. All my questions have been addressed. Thus, I suggest to accept this paper.

---

### Author Response · Authors · 2022-11-18
**A Gentle Reminder**

Dear Reviewers

We apologize for any inconvenience that our message may cause in advance.

Again, we would like to thank you for the time you dedicated to reviewing our paper and your valuable comments.  Since the end of discussion period is close and we have not heard back from you yet, we would appreciate if you kindly let us know if we can be of any further assistance in clarifying any issues.

We humbly remain at your disposal.

Thanks a lot again, and with best wishes

Authors

---

### Decision · Program_Chairs · 2023-01-20

**Decision:**

Accept: poster

**Justification For Why Not Higher Score:**

There are related results in existing works, this is not a groundbreaking new concept.

**Justification For Why Not Lower Score:**

see above, all reviewers broadly agree that this paper is a valuable contribution to ICLR.

**Metareview: Summary, Strengths And Weaknesses:**

This paper translates results from approximation theory to neural ODEs for the solution of PDEs. The presentation is clear, and the reviewers all broadly agree that this paper should be accepted.

**Note From Pc:**

if the above contains the word "oral" or "spotlight" please see: "oral" presentation means -> notable-top-5% and "spotlight" means -> notable-top-25%. As stated in our emails, we are disassociating presentation type from AC recommendations